



# Assessing the adequacy of traditional hydrological models for climate change impact studies: A case for long-short-term memory (LSTM) neural networks

Jean-Luc Martel[1], François Brissette[1], Richard Arsenault[1], Richard Turcotte[2], Mariana Castañeda-Gonzalez[1], William Armstrong[1], Edouard Mailhot[2], Jasmine Pelletier-Dumont[2], Gabriel Rondeau-Genesse[3], Louis-Philippe Caron[3]

[1]Hydrology, Climate and Climate Change (HC3) laboratory, École de technologie supérieure, Montreal, Canada, H3C 1K3
[2]Direction principale de l'expertise hydrique (DPEH), Ministère de l'Environnement et de la Lutte contre les changements climatiques, de la Faune et des Parcs (MELCCFP), Quebec, Canada, G1R 5V7
[3]Ouranos, Montreal, Canada, H3A 1B9

*Correspondence to*: Jean-Luc Martel (jean-luc.martel@etsmtl.ca)

**Abstract.** Climate change impact studies are essential for understanding the effects on water resources under changing climate conditions. This paper assesses the effectiveness of Long Short-Term Memory (LSTM) neural networks versus traditional hydrological models for these studies. Traditional hydrological models, which rely on historical climate data and simplified process parameterization, are scrutinized for their capability to accurately predict future hydrological streamflow in scenarios of significant warming. In contrast, LSTM models, known for their ability to learn from extensive sequences of data and capture temporal dependencies, present a viable alternative. This study utilizes a domain of 148 catchments to compare four traditional hydrological models, each calibrated on individual catchments, against two LSTM models. The first LSTM model is trained regionally across the study domain of 148 catchments, while the second incorporates an additional 1,000 catchments at the continental scale, many of which are in climate zones indicative of the future climate within the study domain. The climate sensitivity of all six hydrological models is evaluated using four straightforward climate scenarios (+3°C, +6°C, -20%, and +20% mean annual precipitation), as well as using an ensemble of 22 CMIP6 GCMs under the SSP5-8.5 scenario. Results indicate that LSTM-based models exhibit a different climate sensitivity compared to traditional hydrological models. Furthermore, analyses of precipitation elasticity to streamflow and multiple streamflow simulations on analogue catchments suggest that the continental LSTM model is most suited for climate change impact studies, a conclusion that is also supported by theoretical arguments.



## 1 Introduction

A warming climate has profound cascading impacts affecting the entire biosphere (e.g., Bellard et al., 2012; Jackson, 2021;
Scheffers et al., 2016). The potential influence of an evolving climate are typically assessed through climate change impact
studies. These studies assess the effects of climate change on environmental, economic, and social systems. They cover how
a changing climate affects ecosystems and the weather, and how it ultimately impacts human population and infrastructures.
Impact studies are a critical tool to enable efficient adaptation strategies addressing climate-related challenges.

There are many different ways to conduct impact studies, but the most common approach is to use a modelling chain connecting
General Circulation or Earth System models (GCM for short) to a specific impact model such as a crop (Jägermeyr et al.,
2021), forest fire (Dupuy et al., 2020) or hydrological model (Hagemann et al., 2013; Minville et al., 2008). A climate change
impact study should not only quantify future changes, but also the uncertainty in the projected change (Chen et al., 2011; Clark
et al., 2016; Wilby and Harris, 2006).

To adequately frame climate change impact uncertainty, the importance of incorporating multiple GCM cannot be overstated.
GCM are instrumental in projecting future climate scenarios, yet the inherent uncertainty in their climate sensitivity—defined
as the Equilibrium Climate Sensitivity (ECS), which quantifies the Earth's temperature response to a doubling of carbon
dioxide concentrations ($CO_2$)—presents a significant challenge. Given the variability in the ECS among different GCM
(Hausfather et al., 2022), leveraging a suite of GCM is essential to adequately sample this pivotal source of uncertainty, thereby
enhancing the robustness of climate projections. This has been the norm for many years, as reflected in a multitude of climate
change studies in hydrology and other fields (e.g., Chen et al., 2012; Deb et al., 2018; Martel et al., 2022; Thompson et al.,
2013; Wang et al., 2020).

The climate sensitivity of impact models has comparatively been much less studied, but has nonetheless been shown to be
significant (Brigode et al., 2013; Giuntoli et al., 2018; Kay et al., 2009; Krysanova et al., 2018; Mendoza et al., 2015; Poulin
et al., 2011), sometimes to the point of being more important than that of GCM (Her et al., 2019). The climate sensitivity of
impact models is dependent on the parameterization of various (and often simplified) processes. The calibration parameter sets
are typically optimized for historical climatic conditions (e.g., Althoff and Rodrigues, 2021; Chlumsky et al., 2021) and may
not be well-suited to future climates, especially under scenarios of significant warming. Recognizing and evaluating the
uncertainty tied to impact model sensitivity is crucial and is typically approached by employing multiple impact models or
multiple parameter sets when feasible. However, using multiple impact models, without a priori knowledge of their
transferability to a different climate is a likely path towards an overestimation of impact model uncertainty, which is as likely
to lead to maladaptation as underestimating it (Sem, 2007). Mearns (2010) and many others emphasize the crucial importance
of correctly framing uncertainty to help decision-makers adopt proper adaptation measures.



In hydrology, this has spurred a body of literature focused on refining models and calibration approaches for hydrology models to better account for future climate variability and change. Using physically-based hydrological models (Feng et al., 2023; Michel et al., 2022) or best-performing models (Li et al., 2015) has been proposed as a more robust alternative. However, such models typically require complex observational inputs that are often not available, and even the most physical models do require some level of parameterization. Hydrological models always had to contend with internal climate variability and this is why a calibration period should be as long as possible, as argued by Arsenault et al. (2018) and Shen et al. (2022) for optimal robustness. They suggest that by maximizing the length of the calibration time-series, it exposes the models to more contrasted conditions and therefore improves robustness. However, internal climate variability over a typical calibration historical period remains small compared to end of century climate projections, especially for near-surface temperature. To address this, multi-model approaches (Arsenault et al., 2015; Seiller et al., 2015) and various split-sample procedures have been proposed to study model robustness over contrasting periods, such as dry/wet or cold/hot periods (e.g., Bérubé et al., 2022; Coron et al., 2012; Thirel et al., 2015; Vansteenkiste et al., 2014). Ultimately, none of the above approaches have proven particularly convincing. In particular, Bérubé et al. (2022) conducted a large-sample study of contrasting-conditions calibration strategies, and showed that no single calibration strategy or length was successful for all metrics and study catchments. Some of the underlying reasons for that are discussed by Duethmann et al. (2020). Finally, the large number of studies on regionalization also demonstrated that hydrological models have a relatively limited transferability to other catchments even in similar climate zones (Arsenault and Brissette, 2014b; Guo et al., 2021; Parajka et al., 2013; Tarek et al., 2021).

In this context, deep learning models may have the ability to overcome such problems (Althoff et al., 2021; Wi and Steinschneider, 2022; Zhong et al., 2023). In particular, Long Short-Term Memory (LSTM; Hochreiter and Schmidhuber, 1997) networks offer a promising alternative. LSTM models are a special kind of Recurrent Neural Network (RNN) architecture which can learn from sequences of data by capturing temporal dependencies and relationships. They are specifically designed to avoid the long-term dependency problem of vanishing or exploding gradients during training. Their unique architecture enables them to learn and remember over long sequences of data, making them highly effective for predictions of time series. In addition, unlike traditional conceptual models that are typically calibrated on data from a single catchment or from a small number of catchments pooled together (Gaborit et al. 2015, Ricard et al. 2012), LSTM models are trained across a diverse array of catchments, encompassing a wide range of climatic conditions and physical characteristics, potentially covering a range similar or beyond that expected due to climate change over many catchments. For these reasons, this methodological shift is anticipated to yield models with enhanced robustness to varying climate scenarios. Kratzert et al. (2019a); Kratzert et al. (2019b) underscored this potential, demonstrating that a regional LSTM model can significantly outperform traditional hydrological model regionalization methods, which rely on locally calibrated models. This was then validated on independent datasets by Arsenault et al. (2023), Li et al. (2022) and Nogueira Filho et al. (2022). Kratzert et al. (2024) provided a rationale on why LSTM-based hydrological models should always use more than one catchment for training.



Essentially, deep learning approaches require a large amount of data to be trained properly, and including more data allows the model to better detect patterns and relationships between catchment descriptors, meteorological forcings and the target streamflow.

The implementation of LSTM-based regional hydrological models is an alternative to traditional "trading space for time" 100    methodologies. Trading space for time is an approach used in ecological and environmental studies to infer long-term environmental changes by examining spatial gradients at a single time point. In the context of climate change, this method assumes that spatial variations across different geographical regions can serve as proxies for temporal changes, thereby allowing researchers to predict the effects of climate change over time by observing current spatial patterns (Singh et al., 2014). Using LSTM models for hydrological simulation under changing climate conditions could likewise be compared to methods 105    based on climatic analogues. Analogues-based methods, which identify past weather patterns similar to those projected for the future, offer an intuitive way to understand potential climate impacts by drawing direct parallels with historical events (e.g., Ford et al., 2010; Ramírez Villegas et al., 2011). While such methods provide valuable insights, particularly in elucidating the practical implications of climate projections, they inherently rely on the assumption that past climate variability is a sufficient proxy for future conditions. This assumption may not always hold, especially under scenarios of unprecedented climate change.

The objectives of this paper are threefold. The first is to assess the performance of an LSTM-based hydrological model in a climate change impact study, focusing on its potential ability at capturing future hydrological streamflow. The second is to compare the future streamflow projections derived from the LSTM-based model against those obtained from conventional hydrological models, aiming to identify differences in the response across a spectrum of streamflow metrics and multiple 115    catchments. Finally, the third objective is to explore the climate sensitivity of the LSTM-based model in contrast to traditional hydrological models, thereby contributing to a deeper understanding of LSTM-based hydrological model uncertainties in climate impact studies.

## 2 Study area and data

### 2.1 Study area

This study focuses on a collection of 148 catchments located in the northeastern region of North America. These catchments are characterized by their exposure to snow-related processes, including accumulation and melt phases, playing a significant role in their hydrological dynamics. The selection of these catchments was done through the comprehensive HYSETS database (Arsenault et al., 2020), which catalogues over 14,425 North American catchments, complete with hydrological, meteorological, and geophysical data. The choice of this specific subset was motivated by a previous study in which the same 125    catchments were used in the context of predicting streamflow in ungauged basins (Arsenault et al., 2023). The LSTM models in that study proved to outperform conceptual hydrological models for this task, paving the way to the present study to



determine how regional LSTM models can integrate spatially diverse information to predict streamflow in changing conditions. This is akin to a "trading space for time" approach using the LSTM model to do the work. This diversity is particularly pronounced between the southern and northern catchments, with notable differences in peak streamflow timings and precipitation rates, necessitating a detailed modelling approach beyond simple area-based extrapolations. In the Arsenault et al. (2023) study, only those catchments with a drainage area exceeding 500 km$^2$ were included, thereby sidestepping potential issues related to scale and time-lag in model regionalization efforts. Catchments also required at least 30 years of data over the 1979-2018 period to be selected in order to have sufficient data to train both the conceptual hydrological models and the deep-learning implementations.

For one scenario in this study, an extra set of 1,000 donor catchments was selected. This was done to determine if adding information from more catchments with different climate and physical characteristics could help increase the regionalization ability of the LSTM models, and in turn help increase reliability in terms of climate change impact studies. This was performed by first widening the spatial extents of the study area and pre-selecting catchments with more than 20 years of available streamflow data within the new spatial bounds, as shown in Figure 1a and 1b. From there, 1,000 donor catchments on top of the initial 148 were selected at random to be included in the extended LSTM model's training, with a larger distribution of these catchments being from more southern regions of the United-States. This was done to ensure warmer catchments would be included in the training of the extended LSTM model, aiming to improve its ability to simulate a warmer climate in the northeastern North America catchments.

**2.2 Data**

**2.2.1 Meteorological and hydrometric data**

All hydrological models in this study shared the same meteorological datasets to ensure a fair comparison between models and model types. Indeed, while conceptual models are limited in their type of meteorological inputs, deep learning models can ingest any type of data and extract useful information if it is available. Therefore, the dataset that was the common denominator for all models was used for all models. This consists of maximum and minimum daily temperature, as well as daily rainfall and snowfall. These data were initially provided through the ERA5 reanalysis dataset (Hersbach et al., 2020), but were directly used from the HYSETS database as they were already catchment-averaged and processed at the daily scale for all catchments in this study. Tarek et al. (2020) showed that ERA5 data can be used for hydrological modelling applications in replacement of observed datasets with little to no loss in performance, while ensuring no missing data for the entire period. Meteorological data therefore covered the period 1980-2018 inclusively.



Hydrometric data was taken from the HYSETS database as well, and covered the same period as the meteorological data. However, observed streamflow records contain many missing data, which justifies the use of catchments that only had at least 20 years of observed streamflow in the catchment selection.

Boxplots will be used throughout this paper to outline study results. A boxplot is a concise graphical tool which highlights the central tendency, variability, and outliers within the distribution of results across all of the study catchments. It summarizes the data distribution using a five-number summary: the minimum, first quartile (Q1), median, third quartile (Q3), and maximum values. The boxes stretch from Q1 to Q3, representing the interquartile range (IQR), with a horizontal line at the median. The whiskers extend from the box to the highest and lowest values within 1.5 times the IQR from the Q1 and Q3, marking the data range, while the points outside the whiskers indicate outliers (if present).

Figure 1 presents a first comparison between the meteorological data of the initial 148 catchment set (regional dataset) and the 1,000-catchment extension (continental dataset) using maps and boxplots. Specifically, the regional dataset encompasses a narrower climatic range, with mean annual temperatures varying from 0.5 °C to 11.1 °C and precipitation levels spanning 809 mm to 1425 mm. Conversely, the continental group dataset extends these boundaries significantly, covering a broader spectrum of climate conditions. This dataset records mean annual temperatures ranging from -9.1 °C to 21.8 °C and total precipitation ranging from 328 mm to 1570 mm.





**Figure 1: Maps (a, b) of study area showing the location of the 148 studied catchments (large circles with black outline), and the 1,000 donor catchments for the continental LSTM model (small circles with grey outline). The fill colour represents the mean annual temperature (a) and total precipitation (b) of each catchment. The circles are located at the centroid of each catchment. Box plots showing the comparison of mean annual temperature (c) and total precipitation (d) across the target sample of 148 catchments (green boxes) and the 1,000 donor catchments (orange boxes) for the continental LSTM simulation.**

Such a wide range of key climatic variables enables a comprehnsive assessment of climate impacts across a wider geographic area. The extended range of the continental dataset is particularly critical for the development of robust LSTM models. By incorporating a broader spectrum of mean annual temperatures and precipitation levels, the continental dataset not only captures significant variability within climate data but also enhances the model's capacity to generalize across a diverse array of climate conditions. This aspect is especially beneficial for anticipating and adapting to a future warmer climate, where the variability and extremities of climate conditions are expected to intensify.



### 2.2.2 Catchment descriptors

Catchment descriptors are required for regional LSTM-based hydrological modelling, as the simulated streamflow is a function

of not only the meteorological data, but also the catchment properties. These descriptors allow the LSTM models to learn patterns and relationships to modulate and adjust simulated streamflow based on each catchment's static properties. This has already been implemented in Kratzert et al. (2019a) and Arsenault et al. (2023). The catchment descriptors used in this study represent geographic (i.e., catchment's centroid latitude and longitude, drainage area, elevation, slope, aspect, perimeter and Gravelius index), land-use (i.e., fraction of crops, forests, grass, shrubs, water, wetlands, and urban areas), geologic (i.e.,

permeability and soil porosity), and climatic (i.e., mean total precipitation, mean potential evapotranspiration, mean snow water equivalent, meteorological aridity, snow fraction, frequency of high- and low-precipitation events, as well as duration of high- and low-precipitation events) descriptors, for a total of 26 descriptors. These are the same as those used successfully in Arsenault et al. (2023) and are presented in Figure S1.

### 2.3 Hydrological models

### 2.3.1 Traditional hydrological model setup

The traditional hydrological models are characterized by their lumped and conceptual nature, enabling local calibration across the large array of catchments used in this study. Meteorological data from all ERA5 grid points within the drainage area boundary of each catchment were averaged due to the lumped structure of the models. A total of four traditional hydrological models with a relatively wide range of potential evapotranspiration (PET) estimation methods and degree-day snow models

were used as a benchmark for comparison with the LSTM-based models:

1) GR4J (French for Model of Rural Engineering with 4 parameters Daily) is a parsimonious 4-parameter lumped model developed by Perrin et al. (2003). Due to its limitation in simulating snow processes, it has been coupled with a 2-parameters variant of the "simple as possible, but not simpler" degree-day snow model CemaNeige proposed by

(Valéry et al., 2014), thereby ensuring a basic representation of the evolution of the snow cover. This integration results in a hydrological model termed GR4J_CN, which comprises a total of 6 parameters. PET was computed using the Oudin formula (Oudin et al., 2005), which is a variant from the Mcguinness and Bordne (1972) that showed the best performance among 27 other PET formulas for the simulation of streamflow. Previous studies have demonstrated the effectiveness of this model structure in accurately simulating continuous daily streamflow for snowmelt-

dominated catchments similar to those used in the regional dataset of this study (Troin et al., 2015; Troin et al., 2018).

2) HMETS (Hydrological Model - École de technologie supérieure; Martel et al., 2017), a 21-parameter lumped hydrological model, stands as a simple model originally designed for research and educational purposes. One notable feature of this model is its snow model based on the work of Vehviläinen (1992), a 10-parameter degree-day model which enables the representation of the snowpack's melting and refreezing process. The relatively large number of



parameters allows it to provide good performance on a wide variety of catchments across North America as shown
       by Martel et al. (2017). Similar to GR4J_CN, PET was provided to the model by the Oudin formula (Oudin et al.,
       2005).

   3)  HSAMI, a 23-parameter lumped model, which has been utilized for daily streamflow forecasting across over 100
       catchments by Hydro-Québec, a prominent hydropower producer. A simple empirical PET formula based on

minimum and maximum temperature is used by HSAMI:

$$PET = 0{,}0029718 \cdot (T_{max} - T_{min}) \cdot exp\{0{,}0342 \cdot (T_{max} + T_{min}) + 1{,}216\} \qquad (1)$$

       where temperatures are in °C and PET in cm/day. The model uses a 6-parameter degree-day snow model allowing to

simulate the processes linked with accumulation of snow, rain interception, and melting of the snow cover. HSAMI
       has found application in various hydrological and climate change impact studies, such as those conducted by Minville
       et al. (2008), Arsenault and Brissette (2014a), and Martel et al. (2020).

   4)  MOHYSE, a French abbreviation for "HYdrological MOdel simplified to the EXtreme", is a very basic 10-parameter
       lumped hydrological model created by Fortin and Turcotte (2007) for teaching undergraduates. Despite its simplicity,

it is widely used in research due to its effectiveness in simulating streamflow. The PET estimation method used by
       MOHYSE is inspired by a simplified version of the method proposed by Hamon (1961), and its snow model uses a
       simple 2-parameter degree-day approach. A comparative analysis with the three other lumped models used in this
       study (i.e., GR4J_CN, HMETS and HSAMI) on 3,375 North American catchments (a subset of the HYSETS
       database) showed MOHYSE's performance was lower but still acceptable performance. The model is kept to better

study model structural uncertainty and climate sensitivity,

The HMETS and HSAMI models were calibrated using the Covariance Matrix Adaptation - Evolution Strategy (CMA-ES;
Hansen and Ostermeier, 2001) stochastic optimization method, known for its superior performance in handling larger
parameter spaces (Arsenault et al., 2014). With respect to GR4J_CN and MOHYSE, their calibration was conducted using the
Shuffled Complex Evolution - University of Arizona (SCE-UA; Duan et al., 1992; Duan et al., 1994) optimization method,
which is more suitable for models with smaller parameter spaces (Arsenault et al., 2014). Following the recommendation of
Arsenault et al. (2018), calibration utilized much of the available observations (i.e., data between 1981 and 2007) rather than
the traditional split-sample validation, providing more suitable parameters for climate change impact studies. However, a short
validation period of five years (2008 to 2012) was still kept to allow for a fair comparison between the traditional hydrological
models and the LSTM-based models. A warm-up period of two years was used to ensure reasonable starting values for the
models' state variables. A total of 10,000 model evaluations were performed on each catchment using a modified version of
the Kling-Gupta Efficiency (KGE; Gupta et al., 2009) proposed by Kling et al. (2012), an objective function based on the
correlation (r), variability bias, and mean bias:



$$KGE = 1 - \sqrt{(r-1)^2 + \left(\frac{\sigma_{sim}/u_{sim}}{\sigma_{obs}/u_{obs}} - 1\right)^2 + \left(\frac{\mu_{sim}}{\mu_{obs}} - 1\right)^2} \qquad (2)$$

where $\sigma$ represents the variance, and u the average of the simulation and observed streamflow respectively.

### 2.3.2 LSTM-based model setup

The conceptual lumped hydrological models are compared against an implementation of a Long Short-Term Memory (LSTM) model. LSTM models have been used in many applications related to hydrology, from simple rainfall-runoff modelling in single catchments and on regional domains (Kratzert et al., 2018; Kratzert et al., 2019a), in streamflow forecasting (Girihagama et al., 2022; Sabzipour et al., 2023) and in streamflow prediction at ungauged sites (Arsenault et al., 2023) to name a few. The LSTM model is designed to integrate both dynamic and static features, capturing the temporal patterns of weather variables (e.g., precipitation and temperature) and the intrinsic characteristics of catchments (e.g., drainage area, slope and land-use). The model structure is detailed in the supplementary materials (Figure S2), but it is important to note that the model was implemented twice: Once using a regional set of catchments (regional model in Table 1) and the other integrating data from the extra set of 1,000 donor catchments to improve training, referred to as the continental model in Table 1. For both applications, only the amount of input data for training was changed. The structure and hyperparameters remained exactly the same in both instances. A summary of the LSTM model is presented here.

First, the LSTM model ingests data for four dynamic (i.e., time-series) variables, namely minimum and maximum daily temperature, as well as rainfall and snowfall. For each streamflow simulation day, a 365-day look-back window of previous meteorological data is used to allow the LSTM model to determine the impact of these data on the streamflow for the simulation day. This block of 365 days × 4 variables is then passed to four LSTM layers each having 256 units. Results are concatenated in two branches and then merged with the static data representing the catchment descriptors. These being static, they are represented by a vector in which each element represents a catchment descriptor. The descriptors are passed into a 128-unit dense layer with a Rectified Linear Unit (ReLU; Agarap, 2018) activation layer. A series of LSTM layers and concatenations is then applied to mimic a part of a ResNET network (He et al., 2016; Sarwinda et al., 2021), where shortcuts exist between certain branches. This has led to significant performance in other fields, although for applications with much more data and larger LSTM models. Dropout layers are also added throughout the model to increase its robustness and generalizability, given that it is used as a regional model that should consider a wide array of catchments and hydrometeorological conditions for application in climate change scenarios. The final layers in the model are a series of dense layers and activation functions leading to a single output, which is the estimate of the streamflow for the given inputs.






The model was then trained using the AdamW optimizer (Loshchilov and Hutter, 2017) with a learning rate that was allowed to change according to the model validation performance, using the "Reduce Learning Rate on Plateau" (or RLRP) algorithm (Smith and Topin, 2019). To do so, the model compared the simulated streamflow obtained in training with the observed streamflow for each catchment. However, since data from multiple catchments was combined in the computation of the

objective function, a pre-processing step was implemented. Indeed, the observed streamflow were first divided by the catchment area for each catchment in the dataset, normalizing the streamflow. Then, the Nash-Sutcliffe Efficiency (NSE; Nash and Sutcliffe, 1970), was used as an objective function, which was modified to weigh the NSE values for each catchment according to their observed streamflow deviations, as was done by Kratzert et al. (2019b) and repeated with success on the same 148 catchments as in this present study in Arsenault et al. (2023).


The optimization was performed using data from 1981-2002 inclusively (22 years) for training, from 2003 to 2007 inclusively (5 years) for validation and 2008 to 2012 (5 years) for testing. This allowed providing sufficient training data for both the regional and extended LSTM models, while allowing enough independent data for comparison and evaluation. It is important to note that the validation period in deep learning is not the same as the validation period for conceptual hydrological models.

In deep learning, validation refers to the intermediate evaluation between training epochs and is used as a stopping criterion for model training. When the validation score stops improving or starts deteriorating for a certain number of consecutive epochs, the model stops training and returns the version with the best validation score. It is then evaluated on the testing period, which is the same as the validation period for conceptual hydrological models. The list of traditional conceptual lumped and LSTM-based hydrological models used in this study are summarized in Table 1.


**Table 1: List of hydrological models used in this study.**

| Acronym | Model type | Number of adjustable parameters | Calibration |
|---------|-----------|--------------------------------|-------------|
| GR4J_CN | Lumped conceptual | 6 | Local |
| HMETS | Lumped conceptual | 21 | Local |
| HSAMI | Lumped conceptual | 23 | Local |
| MOHYSE | Lumped conceptual | 10 | Local |
| LSTM-R | Deep learning | - | Regional |
| LSTM-C | Deep learning | - | Continental |

**2.4 Climate change impacts on hydrological simulations**

Two different tests were implemented to evaluate the ability of the conceptual and LSTM-based hydrological models to

simulate streamflow in conditions that differ from those in the historical period. The first is a simple sensitivity analysis in which simple delta factors are applied to historical meteorological data. The second uses the more realistic approach of driving the models with bias-corrected climate model simulations. Both methods are presented in this section.



### 2.4.1 Simple climate sensitivity analysis

A key component of the climate change impacts assessment methodology involved conducting a sensitivity analysis to evaluate
the impact of hypothetical changes in key climatic variables on streamflow within catchments. This method was designed to evaluate how the conceptual and LSTM hydrological models react to these simple but significant changes in meteorological variables. To achieve this, historical weather data was modified by applying predetermined factors to create new datasets that served as rough estimates of future weather conditions. This approach enabled directly assessing the sensitivity of the hydrological system to specific changes in temperature and precipitation, irrespective of the complex dynamics captured by
climate models, as will be explored in the next section.

The sensitivity analysis included four distinct tests. Two were performed by modifying temperature (i.e., +3 °C and +6 °C), reflecting potential increases in daily minimum and maximum temperatures. These adjustments were based on the premise that elevated temperatures can significantly impact evapotranspiration rates, snowmelt timing, and ultimately, streamflow
patterns in catchments. Then, two other tests focused on precipitation (i.e., +20% and -20%), recognizing that climate change could increase or decrease precipitation rates depending on the catchment locations and thus alter streamflow peaks and volumes.

### 2.4.2 Climate models

The second climate change analysis was performed using GCM climate model data to evaluate the differences between
conceptual lumped and LSTM-based hydrological models for simulating more complex scenarios than those generated in the sensitivity analysis.

### 2.4.2.1 Climate model data

To drive the hydrological models (both conceptual and the LSTM models) with future climate data, GCMs were used. For this purpose, output data from 22 GCMs were downloaded from the Coupled Model Intercomparison Project Phase 6 (CMIP6;
Eyring et al., 2016) as shown in Table 2. Availability of the required data was the only factor during model selection. This ensured a broad representation of the current state-of-the-art in climate modelling without any pre-selection bias. As with the historical period simulations, the variables required were maximum and minimum temperature as well as solid and liquid precipitation. This wide array of climate models plays an important role in capturing the range of possible future climate conditions, which allows assessing the robustness of the LSTM-based model compared to the conceptual hydrological models
in various future conditions.




**Table 2: List of the 22 CMIP6 GCM models used in this study.**

| Acronym | Modelling centre | CMIP6 model name | Spatial resolution (degrees, lat. x lon.) | ID number in figures |
|---|---|---|---|---|
| BCC | Beijing Climate Center, China Meteorological Administration | BCC-CSM2-MR | 1.125 x 1.125 | 1 |
| CAS | Chinese Academy of Sciences, Institute of Atmospheric Physics, China | FGOALS-g3 | 2.0 x 2.25 | 2 |
| CCMA | Canadian Centre for Climate Modelling and Analysis, Canada | CanESM5 | 2.8 x 2.8 | 3 |
| CSIRO | Commonwealth Scientific and Industrial Research Organization, Australia | ACCESS-ESM1-5 | 1.125 x 1.875 | 4 |
| EC | EC-Earth Consortium, Europe | EC-Earth3 | 0.7 x 0.7 | 5 |
| | | EC-Earth3-CC | 0.7 x 0.7 | 6 |
| | | EC-Earth3-Veg | 0.7 x 0.7 | 7 |
| | | EC-Earth3-Veg-LR | 1.125 x 1.125 | 8 |
| GFDL | NOAA Geophysical Fluid Dynamics Laboratory, USA | GFDL-CM4 | 2.0 x 2.5 | 9 |
| | | GFDL-CM4 | 1.0 x 1.0 | 10 |
| | | GFDL-ESM4 | 1.0 x 1.0 | 11 |
| INM | Russian institute for Numerical Mathematics | INM-CM4-8 | 1.5 x 2.0 | 12 |
| | | INM-CM5-0 | 1.5 x 2.0 | 13 |
| IPSL | Institut Pierre Simon Laplace, France | IPSL-CM6A-LR | 1.25 x 2.5 | 14 |
| JAMSTEC | JAMSTEC, AORI, NIES, R-CCS, Japan | MIROC6 | 1.4 x 1.4 | 15 |
| KIOST | Korea Institute of Ocean Science and Technology, South Korea | KIOST-ESM | 1.875 x 1.875 | 16 |
| MPI | Max Planck Institute for Meteorology, Germany | MPI-ESM1-2-LR | 1.875 x 1.875 | 17 |
| | | MPI-ESM1-2-HR | 0.94 x 0.94 | 18 |
| MRI | Meteorological Research Institute, Japan | MRI-ESM2-0 | 1.875 x 1.875 | 19 |
| NCC | Norwegian Climate Centre, Norway | NorESM2-LM | 1.875 x 2.5 | 20 |
| | | NorESM2-MM | 0.9375 x 1.25 | 21 |
| NUIST | Nanjing University of Information Science and Technology, China | NESM3 | 1.875 x 1.875 | 22 |


Future climate forcings are those of the Shared Socioeconomic Pathway 8.5 (SSP5-8.5; Gidden et al., 2019), a scenario characterized by high greenhouse gas emissions and significant global warming. While acknowledging that SSP5-8.5 represents a pessimistic outlook on future climate change, this scenario was chosen for its utility in maximizing projected climatic changes. This approach is strategic, aiming to reduce the influence of internal climate variability and enhance the

discernibility of differences between LSTM-based hydrological simulations and those generated by traditional hydrology models. The integration of scenarios with more pronounced changes aligns with the objective to evaluate the adaptability and predictive power of LSTM models in extreme future conditions.



### 2.4.2.2 Climate model data processing

The climate model output data were first processed by applying a bias-correction technique to adjust their outputs. The chosen
method, the Multivariate Bias Correction (MBCn) developed by Cannon (2018), stands out for its efficacy in correcting biases
across multiple meteorological variables simultaneously. This high-performance correction technique ensures that climate
projections maintain statistical properties consistent with the reference datasets, thereby enhancing the reliability of the
hydrological assessments. For this study, the reference data were the same as those used for hydrological modelling (i.e., the
ERA5 reanalysis data). However, no downscaling of climate data was performed, a decision underpinned by the spatial scale
of our hydrological analysis. By aggregating precipitation and temperature data at the catchment scale, we effectively mitigate
potential mismatches between the coarse resolution of CMIP6 GCM grids and the finer scales of the catchments. This approach
is deemed sufficient despite a portion of the catchments being smaller than the typical resolution of CMIP6 models.

### 2.4.2.3 Climate model evaluation period

The bias-correction and climate change simulations were performed on a reference period (1971-2000) and a future period
(2070-2099). This design allows for a direct comparison of climate impacts on hydrology under current and projected
conditions. However, to accommodate the requisite 2-year warm-up period for the hydrological models, the effective analysis
windows are adjusted to 1973-2000 and 2072-2099. This adjustment ensures that the conceptual model states are adequate for
accurate simulation, covering 28 years within each period.

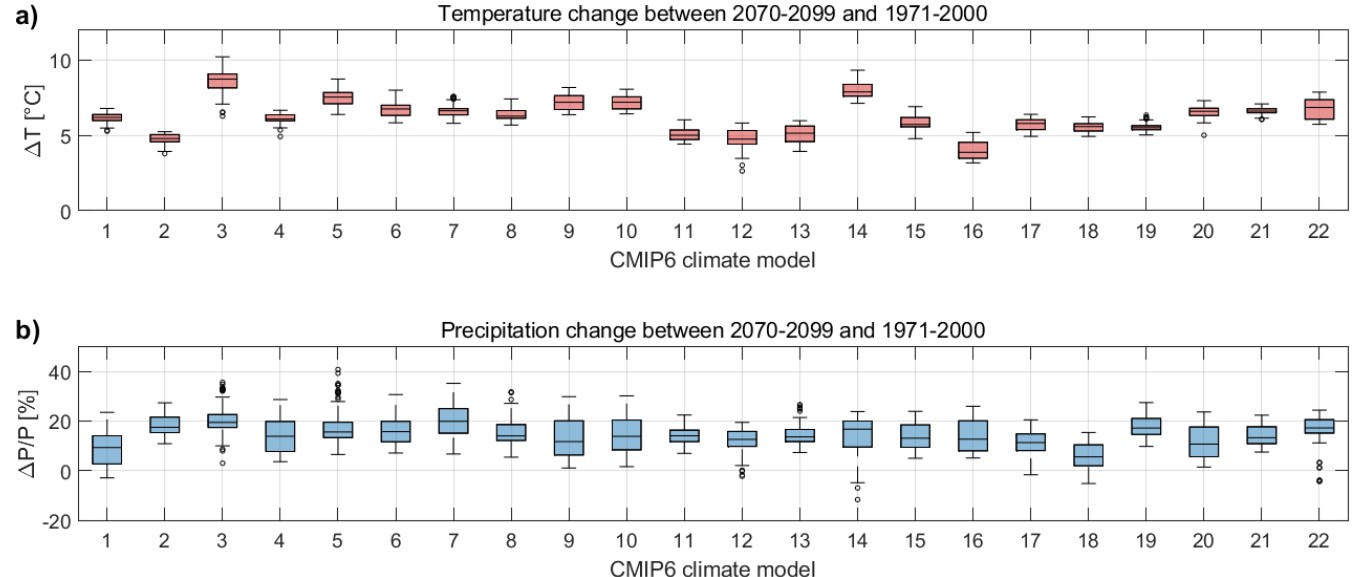


**Figure 2: Distribution of the climate change signal for all 148 catchments for the 22 GCMs for mean annual temperature change
(future - reference; a), and mean annual total precipitation ([future - reference] / reference; b). The future period refers to 2072-
2099, while the reference period refers to 1973-2000, inclusively. Each boxplot contains 148 points (i.e., one per catchment).**





From Figure 2, it can be seen that the climate model projection show a wide range of future conditions, especially in terms of
temperatures (Figure 2-a). CanESM5 (model 3) is the warmest GCM (median increase of 8.7 °C) while KIOST-ESM
(model 16) is the one with the lowest warming (median increase of 3.9 °C). For most models, the range of the increase in
temperature for the various catchments is relatively small, with the widest range being that of CanESM5 (model 3), with
approximately 3.9 °C. This indicates that the temperature variability between models is larger than the spatial variability within
the study domain. In Figure 2-b, it can be seen that precipitation increases are almost universally projected with median
increases from 5.6% in MPI-HR (model 18) to 19.9% for CanESM5 (model 3). For precipitation, the variability between
catchments is much higher than the variability between climate models, at least at the climate time scale.

## 2.5 Evaluation metrics used in this study

In this study, six streamflow metrics were selected to provide a general overview of the hydrological cycle: annual mean
streamflow (QMA), winter (QMDJF), spring (QMMAM), summer (QMJJA), fall (QMSON) mean streamflow, and mean
annual maximum streamflow (QMM). The computed metrics specifically target various aspects of streamflow behaviour:

- Annual metric: to assess the overall hydrological response on a yearly basis, offering insights into long-term changes
  and trends;
- Seasonal metrics: these metrics show the temporal distribution of streamflow, allowing for the identification of shifts
  in hydrological patterns across different times of the year;
- Extreme metric: the evaluation of extremes provides information on potential flooding conditions, which are
  important for risk assessment and adaptation strategies. However, they are also the most likely to show divergences
  between methods due to their de facto rarity in datasets, meaning models have fewer examples to learn from than
  more common streamflow events.

In addition to the streamflow metrics, the KGE and NSE metrics were computed over the historical period for model calibration
and validation (calibration, validation and testing for LSTM models). This allowed comparing the results using metrics that
provide an overall assessment of model fit in terms of bias, correlation and variability.

## 3 Results

### 3.1 Model calibration, validation and testing

This section presents the validation/testing results for both the conceptual and LSTM-based hydrological models. First, Figure
3 presents various performance metrics over the 5-year independent testing period (2008-2012): Kling-Gupta Efficiency
(KGE), Nash-Sutcliffe Efficiency (NSE), relative bias ($\beta$), correlation coefficient ($r$), variance ratio ($\gamma$), and normalized root
mean square error (NRSME). Figure S3 in Supplementary materials presents the results over the calibration period (1983-



2002). Note that the validation period for the LSTM-based models is not shown as there is no comparison for the conceptual

models.



**Figure 3: Kling-Gupta Efficiency (KGE; a), Nash-Sutcliffe Efficiency (NSE; b), relative bias (β; c), correlation coefficient (r; d), variance ratio (ɣ; e), and normalized root mean square error (NRSME; f) metrics over the independent 5-year testing period (2008-**
**2012).**




From Figure 3, it can be seen that the LSTM-based models significantly outperform the conceptual hydrological models for the KGE and NSE metrics. Results between the conceptual models are somewhat similar, with GR4J and HMETS leading the pack and HSAMI not far behind in third place. MOHYSE, on the other hand, displays the lowest performance although the median KGE and NSE are still above 0.7 and 0.5 respectively. The LSTM model variants display better scores, with the continental model (LSTM-C) performing better than the regional one. In terms of NSE, the LSTM-C shows a median value of 0.77, whereas the best conceptual model obtains a median value of 0.63. The regional LSTM has a value slightly above 0.70 for comparison. KGE values for the LSTM-C are again significantly better than those of the nearest contender (LSTM-R, 0.78) or the best conceptual model (HMETS, 0.76). Overall, these results indicate that the models were able to simulate streamflow adequately on the 148 catchments and could be used for the remainder of this study, save perhaps the MOHYSE model, whose impacts will be considered in light of these validation results.

To further investigate the source of the gains made by the LSTM models for the KGE metric, Figure 3c to f presents the KGE results for the independent verification period as decomposed into their individual elements, i.e., relative bias, correlation coefficient and variability ratio. The normalized Root Mean Square Error (NRMSE) was also computed, by which the RMSE was divided by the streamflow range. It can be seen that the relative bias (Figure 3c) is similar between models with more than 75% of the modelled catchments having a negative bias. The LSTM-R model shows a slightly larger negative bias than the others, and the LSTM-C has the lowest one. The correlation coefficient, on the other hand, clearly shows that MOHYSE lost much of its performance due to its poor correlation coefficient, but that both LSTM models scored much higher than the other models for this metric (Figure 3b). LSTM-C had a particularly large correlation coefficient, with a median correlation coefficient of almost 0.9, much higher than the strongest conceptual model with approximately 0.82. Interestingly, the variance ratio (Figure 3e) shows a striking difference between the conceptual and LSTM-based models. Indeed, the conceptual models tend to overestimate the variability (median values larger than 1) and LSTM-based models tend to underestimate it (median values below 1). In both cases, the over/under-estimation of the median is approximately 10%. Finally, the NRMSE (Figure 3d) shows a slightly smaller relative error for the LSTM models, especially the LSTM-C model. However, differences are not as striking as for correlation or variance ratio. In summary, compared to the traditional hydrological models, the LSTM models have similar bias, slightly lower variability but a much stronger correlation, which suggest a better streamflow timing performance.

## 3.2 Results of the sensitivity analysis

Figures 4, 5 and 6 show the expected changes for mean annual (QMA), summer (QJJA) and fall (QSON) streamflow for the four sensitivity cases. Results for the three other metrics (QMDJF, QMAM and QMM) are presented in supplementary materials, Figures S4-S6. In all four scenarios, the four conceptual models behave in a similar way, with small differences





depending on the model structure and scenario. However, some divergence is observed in the response of traditional versus
LSTM-based models.

The most notable difference is the lower sensitivity of LSTM models to changes in precipitation. When precipitation is altered
by adding or subtracting 20%, the LSTM models show smaller changes for all six metrics (Figures 4 and 5 and S4 to S6). This
is particularly the case for the continental LSTM. The median mean annual streamflow change for a 20% increase in
precipitation is 27.9% for LSTM-C compared to 35.5% (GR4J) 34.33% (HMETS) 33.1% (HSAMI) and 31.1% for MOHYSE,
with a median average of 33.5%. The LSTM models may incorporate a more nuanced understanding of the hydrological
balance compared to the conceptual models, whose change in streamflow is larger than that of the LSTM models for the same
variation in precipitation. However, although the results of the LSTM models are different from those of the conceptual models,
it is still unclear which of these are more representative of real-world impacts of climate change.


The differences are more nuanced for the temperature increases. For QMA (Figure 4), both LSTM models are more sensitive
than the 4 traditional models and predict a larger decrease in mean annual streamflow for the +3 °C and +6 °C scenarios. For
summer streamflow (Figure 5), LSTM-C, along with HSAMI are the most sensitive models. The largest differences are seen
for fall flows (Figure 6), where both LSTM models predict streamflow decreases twice as large as that of traditional models.


Despite some variability between the different streamflow metrics, LSTM models appear to be more sensitive to temperature
increases, and less sensitive to precipitation changes. They predict more substantial decreases in mean annual streamflow as
temperatures rise. The LSTM models' pronounced response to a warming climate—especially the marked streamflow
reductions under the 6 °C increase scenario—underscores their potential for providing more accurate projections in regions
expected to experience significant temperature rises.







**Figure 4: Projected mean annual streamflow (QMA) changes for the 4 sensitivity scenarios: temperature increase of +3 °C (a) and +6 °C (b) and precipitation relative change of -20% (c) and +20% (d).**






Figure 5: Same as Figure 4, but for projected mean summer (JJA) streamflow changes.





Figure 6: Same as Figure 4, but for projected mean fall (SON) streamflow changes.

## 3.3 Results of the climate change impact study

The sensitivity analysis provided some insights about the traditional and LSTM hydrological models sensitivity to temperature increase and precipitation changes. From Figure 2, it is clear that the SSP5-8.5 scenario corresponds more closely to a combination of the +6 °C and +20% precipitation sensitivity scenarios. Figure 2 also shows that the warmest models also tend to be the wettest, which follows the fundamental principles of atmospheric physics and thermodynamics, with increased evaporation and convection in a warmer climate. The observed increased sensitivity to temperature of the LSTM models will therefore be moderated by their lower sensitivity to precipitation increases, and one might expect that all 6 hydrological models may fall on a more or less similar baseline. In addition, the simple sensitivity scenarios did not alter the annual cycles of



precipitation and temperature, whereas GCM-derived scenarios present more realistic, but also more complex future projections.

Figure 7 presents combined projected changes for all 6 streamflow metrics and 6 hydrological models. Each boxplot represents the distribution of 3,256 values, one for each combination of 148 catchment and 22 GCMs. Results show that LSTM models tend to behave differently than the four traditional hydrological models, and in a manner consistent with some of the observations made from the simple sensitivity studies. In particular, we can note the following features:

- For mean annual streamflow (QMA), LSTM models project a small median decrease whereas traditional models see a small median increase;
- LSTM models tend to predict lower streamflow and particularly for the fall season (QMSON);
- LSTM models tend to predict larger streamflow during spring (MAM).

Overall, these relative behaviour between traditional and LSTM hydrological models is qualitatively similar to the results of the +6 °C sensitivity analysis: Figure 7 is similar to the upper right panel (+6 °C scenario) of Figures 4 to 6, and Figures S4 to S6. This suggests that in the balance of increasing temperature and precipitation displayed by most climate models, temperature increases play a more significant role in the hydrological cycle.





**Figure 7: Boxplots of the range of expected change for all 6 streamflow metrics. The boxplots are aggregations of the 22 GCMs and 148 catchments. The boxplots are thus drawn from a distribution of 3,246 points (148 catchments × 22 GCMs).**





One objective of this study is to look at the climate sensitivity of hydrological models, which is how their future streamflow response depends on forcing-induced changes in climate variables near the surface. In order to do so, we have looked at the

differential response of all 6 hydrological models to the following combinations of 6 contrasted climate models: hot vs cold, high vs low precipitation scaling, wet vs cold models. Precipitation scaling evaluates the sensitivity of precipitation change to an increase to temperature. Here, precipitation scaling is simply the increase % in mean annual precipitation divided by the mean annual temperature increase. The median changes between both pairs of three climate models are outlined below:

1) Hot vs cold climate models:

- The 3 hottest models are CanESM5 (model 3: +8.75 °C), IPSL (model 14: +7.90 °C) and EC-Earth3 (model 5: +7.55 °C).
- The 3 coldest models are KIOST-ESM (model 16: +3.88 °C), INM-CM$-8 (model 12: +4.76 °C) and FGOALS-g3 (model 2: +4.80 °C).


2) Highest vs lowest precipitation scaling climate models:

- The 3 highest scaling models are FGOALS-g3 (model 2: +3.65%/°C), KIOST-ESM (model 16: +3.28%/°C) and MRI-ESM2-0 (model 19: +3.12%/°C).
- The 3 lowest scaling models are MPI-ESM1-2-LR (model 17: +1.00%/°C), BCC-CSM2-MR (+1.5%/°C) and
NESM3 (model 20: +1.62%/°C).

3) Wet vs dry climate models:

- The 3 wettest models are eC-Earth3-Veg (model 7: +19.9%), CanESM5 (model 3: +19.4%) and FGOALS-g3 (model 2: +17.5%).
- The 3 driest models are MPI-ESM1-2-LR (model 17: +5.6%), BCC-CSM2-MR (model 1: +9.3%) and NESM3 (model 20: +10.7%).

The results are presented in Figure 8 for mean annual streamflow (QMA). Figure 8 shows the changes (in percentage) between both contrasted groups for all 6 hydrological models. For example, for the "hot" vs "cold" model test, a value of -10% indicates that mean annual streamflow is 10% smaller for the three hottest models compared to the 3 coldest ones (corresponding for

example to a 5% increase for the hot models, and a 15% increase for the cold ones). Each boxplot of Figure 8 is therefore made of $148 \times 3$ values, corresponding to the number of catchments and climate models. The median results are shown in Table 3.

Results do not show large differences in terms of annual climate sensitivity. This applies to both the traditional hydrological model group, as well as when comparing the LSTM models to the traditional ones. The LSTM models show a larger sensitivity





to temperature and precipitation scaling (Figure 8 top and middle). The differences are less striking than for the sensitivity analysis and this is likely because hot models also tend to be wet as discussed above.

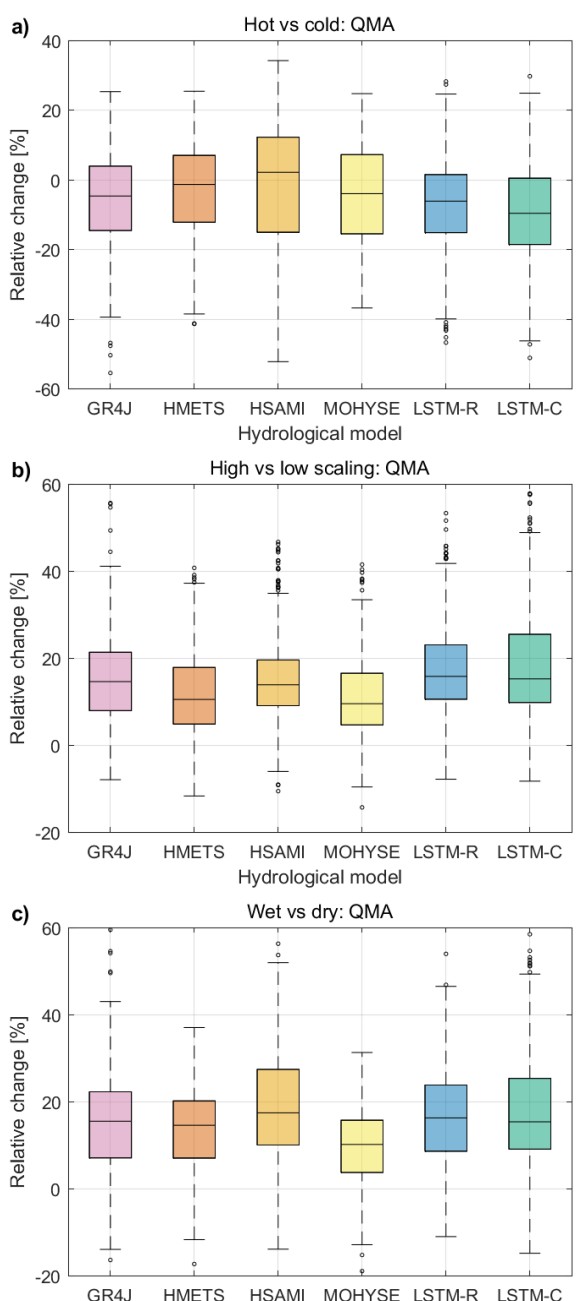

**Figure 8: Boxplots of % change between two groups of three contrasted climate models. a): hot vs cold models, b): high vs low**
**precipitation scaling models, c): wet vs dry models. The boxplots are drawn from a distribution of 444 points (148 catchments × 3 climate models).**




**Table 3: Median results from Figure 8, indicating % change between two groups of three contrasted climate models.**

|          | GR4J  | HMETS | HSAMI | MOHYSE | LSTM-R | LSTM-C | Average Trad. | Average LSTMs |
|----------|-------|-------|-------|--------|--------|--------|---------------|---------------|
| hot/cold | -4.7  | +2.1  | -4.0  | -1.3   | -6.1   | -9.6   | -1.98         | -7.85         |
| high/low | +14.6 | +13.9 | +9.5  | +10.5  | +15.8  | +15.2  | +12.13        | +15.5         |
| wet/dry  | +15.5 | +17.4 | +10.2 | +14.6  | +16.3  | +15.4  | +14.43        | +15.9         |

## 4 Discussion

This section presents the analysis of previous results and interprets them within the context of existing literature.

### 4.1 Calibration and testing

In evaluating the six hydrological models, the testing period (equivalent to the validation period for traditional hydrological models) serves as a common independent period for model intercomparison (refer to Figure 3). Results indicate that LSTM models, particularly LSTM-C, significantly outperform traditional models, as highlighted in recent state-of-the-art papers on LSTM (e.g., Arsenault et al., 2023; Kratzert et al., 2019a; Li et al., 2022), provided they are calibrated with a sufficiently large sample of catchments (Kratzert et al., 2024).

The training period performance of LSTM models (see Figure S3), especially LSTM-C, should not be overly emphasized, impressive though it may be. The numerous parameters in LSTM models can and will lead to overfitting if left unchecked. The validation period, which has no equivalent in traditional hydrological model calibration, acts as the stopping criterion and thus is excluded from both the training and the data scaling/normalization processes. Only training data are used for scaling, and then the parameterized scaling function is applied to other data sets to prevent contamination.

From Figure S3, it is evident that LSTM training period performance surpasses that of conceptual models due to the significantly greater number of degrees of freedom. LSTM training scores, such as NSE/KGE, benefit from much better correlation, though variance ratios are slightly lower compared to those of hydrological models, which almost exactly match the levels of variance. There is a broader spread of bias for LSTM, albeit centered at 0.

Nonetheless, LSTM models demonstrate superior performance during the independent period, indicating no overfitting despite having many more parameters than traditional hydrological models. LSTM-C, benefiting from data from 1,000 additional donor catchments, shows enhanced performance even when tested on the independent period. This indicates the model's ability to leverage additional catchment data effectively. It underscores the necessity for a large sample size to maximize the potential of LSTM models (Kratzert et al., 2024).





LSTMs also excel in utilizing more diverse data types (meteorological and others) than traditional hydrological models. The hydrological models used in this study cannot take advantage of any catchment descriptors for example. More physically-based mpdels could in theory make use of such data. The LSTM implementations discussed in this paper, which only use daily minimum and maximum temperatures, rain, and snow, represent a fraction of their full potential. Incorporating additional variables would likely improve performance further, albeit at the cost of increased training resources and the need for a more flexible model structure with additional nodes to maximize the potential of the added data. This could also enhance the robustness of LSTM models to climate change by constraining outputs more effectively.

However, the complexity of implementing more advanced LSTM models means that all sources of information must also be available for all future time horizons. For instance, employing remote sensing observations (e.g., satellite data) as inputs to LSTM models would likely further improve streamflow simulations for the current period, but such data is not available for future climate change scenarios.

Finally, it should be recognized that not all hydrological models should be considered equal in the interpretation of results. It is quite clear that the continental LSTM model (LSTM-C) clearly outperforms its regional counterpart, from a theoretical and practical point of view. Similarly, the MOHYSE hydrological model is clearly the worst traditional model in this study, from both a performance point of view and based on its simple fully parameterized model structure. The inclusion of MOHYSE and LSTM-R is based on the need to better understand hydrological model sensitivity to a warmer climate.

### 4.2 Sensitivity analysis interpretation

The sensitivity analysis conducted in this study serves as a preliminary approximation, focusing on the expected changes in precipitation and temperature as predicted by GCMs. When looking at Figure 2, it can be seen that most GCMs' median projected precipitation increase is in the 15 to 20% precipitation range, which supports the use of the +20% scenario. Although the -20% scenario may not appear realistic over the study area, its use enhances the overall understanding of the model limitations. This approach enabled assessing how the models would react to reductions in precipitation and thus gain deeper insights into their performance in climate change studies. It is also worthwhile to stress that Figure 2 presents mean projected changes at the annual scale. At the seasonal scale, many GCMs project important decrease in precipitation for certain seasons.

Most median temperature projections among the GCMs ranged between 5 °C to 8 °C increases, hence the choice of the +6 °C. These projections are all from the SSP5-8.5 scenario, which is increasingly seen by many as an overly pessimistic scenario (e.g., Hausfather and Peters, 2020).




By independently altering each variable—precipitation and temperature—we were able to more accurately evaluate the impact of each change, thus avoiding the complication of introducing confounding factors, which is the case when solely relying on GCM simulations for this analysis. This approach provides a clearer understanding of how changes in these variables could impact streamflow, an essential factor in climate change impact assessments. Compared to the hydrological models used in this study, our results clearly show that LSTM models have an increased sensitivity to temperature and a lower sensitivity to

precipitation. These observations are useful in the interpretation of streamflow projections obtained from GCM-derived climate scenarios.

### 4.3 Climate model based impacts assessment

This analysis, based on climate model projections, integrates the temporal dynamics between variables, compared to sensitivity analysis, which preserves the historical timing of events. This more complete picture provides details on transformations in

hydrological indicators under climate change and allows for comparing the conceptual and LSTM-based models under these conditions.

The analysis showed that the results were generally consistent with those of the sensitivity analysis but were not as contrasted, likely due to the combination of precipitation and temperature changes that tend to compensate for one another in the LSTM

models. An important objective of this paper was to investigate the climate sensitivity of hydrological models. The climate sensitivity of traditional hydrological models mostly comes from model structure and complexity, but also from parametric uncertainty. The four traditional hydrological models belong to the same general family of lumped conceptual models. Within that group, model complexity varied significantly, especially with respect to snow and evapotranspiration models, ranging from extremely simple parametric formulas (MOHYSE) to more complex formulations such as the HMETS snow model.

Based on these considerations, it was perhaps surprising to observe that climate sensitivity was relatively similar across all six hydrological models. The LSTM models did show a larger sensitivity to temperature in the hot/cold models experiment. LSTM models also showed a larger sensitivity to high vs. low precipitation scaling, but not when looking at the wet/dry models experiment. All four traditional hydrological models were similar in their climate sensitivity, and even the simple MOHYSE model was not an outlier for the six streamflow metrics considered. It is important to note that the climate sensitivity experiment

only looked at the mean annual streamflow metric. No effort was made to sideline some of the climate models that have high climate sensitivity, also known as "hot models" (e.g., Hausfather et al., 2022; Kreienkamp et al., 2020; Rahimpour Asenjan et al., 2023).

### 4.4 The main question: Which hydrological models should we trust more for climate change impact studies?

This is ultimately the most important question, but also the most difficult one to answer. Since there are no future streamflow

data available, we are mostly left with theoretical arguments. Many studies suggest that we should have more confidence in hydrological models that yield better results in the historical period, as they are better at representing processes (Krysanova et





al., 2018). Based on this sole consideration, LSTM methods should be favored. However, no matter how appealing the argument, relying solely on model performance over the historical record falls short in a few aspects. The concept of the "death of stationarity" (e.g., Galloway, 2011; Milly et al., 2008) tells us that past hydrological behavior is not a reliable indicator of
future conditions, meaning that a model calibrated solely on historical data might not accurately capture future dynamics. As discussed above, all hydrological models are somewhat parameterized, even the most physically based ones, and there is no guarantee that the climate sensitivity of these parameterizations is adequate.

The observation that all six hydrological models display similar climate sensitivity is perhaps comforting but also perhaps
premature, as our analysis did not examine this in much detail, and clearly, a more detailed seasonal analysis looking at more streamflow metrics is warranted. Still, there is an argument to be made that the continental LSTM-C model is the best fit for climate change studies. The inclusion of 1,000 additional catchments, mostly located in a warmer climate, indicates that LSTM-C should have, at the very least, a theoretical advantage. It has learned the complex relationship between climate variables and streamflow, not only over the study domain but also from 1,000 catchments, many of which are representative of, and even
warmer than, the expected end-of-century climate over the study domain. LSTM models are particularly fit at capturing the complex, non-linear climate interactions leading to streamflow. They may, therefore, be better at capturing the temporal dynamics of hydrological processes and their sensitivity to long-term changes in climate patterns. However, LSTM model process representation is non-existent, meaning it cannot be probed directly to assess how the hydrological cycle is modeled. This is a limitation, as it is not possible (or extremely difficult) to assess this in an LSTM-based model, which means it requires
more blind trust than conceptual models.

LSTMs' main weakness is their necessity to ingest large amounts of data. For extreme or rare events, it is likely that the training dataset contains too few of these events, leading to more doubtful performance compared to conceptual models. In such cases, traditional hydrological models may still be better, despite other weaknesses. For example, the parameter sets of conceptual
hydrological models are fixed during the calibration period with the hypothesis that these are constant over time. However, this has been shown to be inaccurate (e.g., Kim et al., 2015; Mendoza et al., 2015; Bérubé et al., 2022). However, beyond theoretical arguments, there are ways to practically investigate hydrological model fitness. Two possible ways are to look at precipitation elasticity of streamflow and to look at climate analogues.

### 4.4.1 Precipitation elasticity of streamflow

One metric that can be used to assess the sensitivity of streamflow to precipitation is the precipitation elasticity of streamflow index, which is defined here as the change in mean annual streamflow divided by the change in mean annual precipitation. For example, an elasticity of 0.5 means that for every change of 1% in precipitation, streamflow responds with a change in the same direction of only 0.5% (Schaake, 1990). Results in Figure 4c-4d showed significant disparities in how both types of hydrological models (conceptual and LSTM-based) handled changes in precipitation volumes. The traditional model median





streamflow elasticity was 25% larger than that of LSTMs. GR4J, the best-performing hydrological model, had 25% of catchments with a streamflow electricity greater than 2 compared to none for LSTM-C. The disparity is quite clear from Figure 9, which shows the spatial distribution of precipitation elasticity for all tested models (traditional and LSTM-based). The observed spatial distribution of elasticities for the LSTM-C model (Figure 9f) is similar to that observed by (Sankarasubramanian et al., 2001) using observed data, with larger elasticity indices south of the Great Lakes.


Figure 9: Rainfall elasticity for each of the 148 catchments as evaluated by the six hydrological models.



Precipitation elasticity can be used as a metric to assess how each model type compares to values obtained from the literature
(Maharjan et al., 2022). This analysis has one major caveat in that it considers the same temporal pattern of precipitation but
with modified amplitude. This is a simplistic representation of the precipitation drivers and processes but, nonetheless, it allows
comparing the obtained values to established precipitation elasticity of streamflow values in the literature to validate plausible
ranges.

Chiew (2006) provide an estimate of precipitation elasticity of streamflow for catchments throughout the world, using the
median elasticity of all available years of data for a set of over 500 catchments worldwide. Overall, their study shows that
elasticity values ranging between 1.0 and 3.0 are reasonable, with values below 2.0 being more representative for colder,
higher-latitude catchments such as those in this study. From Figure 2 in Chiew (2006), it can be seen that most catchments
found within this study's region of interest have elasticities below 1.5, with a few showing values below 1.0. There is one
exception of a catchment having an elasticity of 2.5, and a few between 1.5 and 2.0, but most are below 1.5, lending credence
to the LSTM model estimates (see Figure 4d and Figure 9e-9f). These values of streamflow elasticity also match those obtained
by Sankarasubramanian et al. (2001) in a large sample US study. They show that values above 1.5 are typical of arid/semi-arid
catchments which are not found within our study area. Zhang et al. (2022) recommend to use decadal timesteps to assess the
elasticity, which is more in-line with what was performed in this study.


### 4.4.2 Catchment analogues

One additional way to quantitatively assess the climate change fitness of hydrological models is to examine how the six
hydrological models perform on analogue catchments. Essentially, for the sensitivity analysis (e.g., +6 °C scenario), we look
for catchments within the 1,000-donor group that have similar annual precipitation and temperature cycles but with the latter
being 6 °C warmer. For each of the 148 target catchments, the best 10 analogues out of 1,000 were selected, based on Euclidean
vectors composed of the following values: mean annual temperature, mean annual precipitation, catchment areas, 12 monthly
mean temperatures, and 12 monthly mean total precipitations. It was decided not to use any physical catchment descriptors
(e.g., land cover) as this could possibly favour the LSTM models, which can make explicit use of such descriptors. To find the
best analogues, unequal weights were used to put the emphasis on mean annual temperature (weight equal to 10), precipitation
annual cycle (5) and surface area (10). Several weighting schemes were tested. The weighting schemes had an influence on
the choice of analogues but only a negligible impact on the overall results. Figure 10 shows typical results for catchments 30,
40, and 140. It shows the annual mean hydrographs for all six hydrological models for the +6 °C scenario, superimposed on
the envelope of mean streamflow hydrographs from the 10 closest analogues. All streamflow hydrographs were normalized to
a unit area to account for size mismatch. Figure 10 also shows the target catchment location (red circle) as well as the 10
closest analogues (blue circles).



Figure 10 shows that there is a large disparity in performance among hydrological models, especially in the spring and fall. From this limited sample, we can see that LSTM-C appears to perform the best. Among traditional hydrological models, HSAMI displays the best performance, while MOHYSE and HMETS are the worst. We can also see that the disparity within traditional models appears much larger than that of the two LSTM models.

**Figure 10: Left column: mean annual streamflow hydrographs for catchments 30, 50 and 140. The hydrographs are superimposed over the shaded area of the ten +6 °C closest analogue catchment from the 1,000-donor population. Right column: location of target catchment (red circle) and the 10 closest analogues (blue circles).**





In order to extend the analysis of Figure 10 to all catchments and provide a more quantitative assessment, the root mean square difference between the annual streamflow hydrograph of each hydrological model and the mean of the 10 analogues was computed for all catchments. The results are shown in Figure 11. The results confirm the main observations made from Figure

10 which are that LSTM-C provides the best streamflow estimates for the analogues. HSAMI is clearly second best while MOHYSE and HMETS are the worst. Figure S7 decomposes the results of Figure 11 into the four seasons for a more detailed picture. Furthermore, the results for the same analysis but applied to the +20% precipitation case are presented in Figures S8 (annual scale) and S9 (seasonal scale).

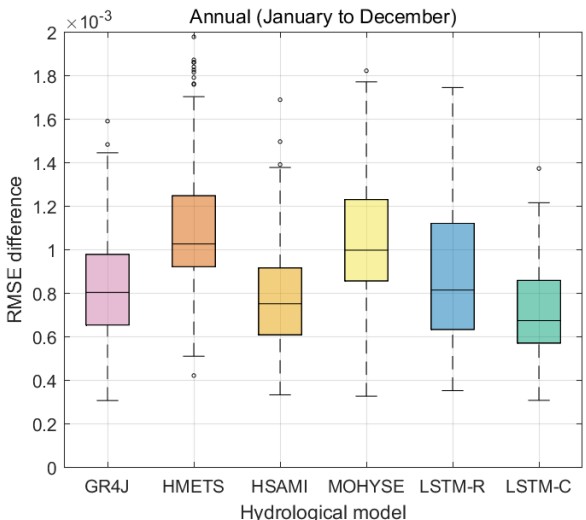


**Figure 11: RMSE difference between the annual mean hydrograph and the mean of the 10 closest +6 °C analogue catchments out of the 1,000-donor population. The boxplots are drawn from the 148 RMSE values, one for each catchment. Seasonal performance is presented in Figure S7.**

Figure 12 presents the spatial distribution of the performance of the six models in reproducing streamflow over the selected analogues. For each of the 148 catchments, it shows the relative performance of each hydrological model on a 1 to 6 ranking, with 1 being the best performance and 6 the worst. We observe that HMETS and MOHYSE are consistently among the worst across the regional domain. These two traditional models, one with a complex and the other with a simple model structure, both indicate that they would be inadequate for climate change studies, or at least not as preferable as the other alternatives,

despite the fact that HMETS performed clearly better than MOHYSE over the historical period. This tends to refute the earlier argument that the better-performing model over the historical period should be favoured. GR4J performs well over the northern domain but does not do as well over the southern portion. Both the regional LSTM-R and HSAMI exhibit different performances over the southern domain. It is interesting to note that the southern domain falls within the Dfa climate zone,





characterized by long, hot, and humid summers, unlike the Dfb zone above. The continental LSTM-C ranks amongst the best
across the entire domain and appears to be the most robust in predicting adequate streamflow for the +6 °C scenario.



**Figure 12: Performance ranking from 1 (best) to 6 (worst) of 6 hydrological models at reproducing the mean streamflow hydrograph**
**from the 10 best +6 °C analogues out of the 1,000-donor catchments.**





The same approach was used to evaluate the +20% precipitation scenario. Results are presented in Figures S8-S10. Figures S8 and S9 respectively present the RMSE difference between the annual mean hydrograph (Figure S8) or seasonal mean hydrograph (Figure S9) and the mean of the 10 closest +20% precipitation analogue catchments out of the 1,000-donor population. Figure S10 presents the performance ranking of models at reproducing the streamflow hydrograph of analogues for the +20% precipitation case (i.e. same as Figure 12 but for the +20% precipitation case). For this scenario, the continental LSTM is once again the best-performing model.

The analogue approach, as used in this study, faces several limitations. It is based solely on precipitation, temperature, and area, without considering other variables. This limitation makes it difficult to find perfect analogues for the study areas. Even small differences in precipitation and temperature, which might seem negligible when selecting analogues, can lead to significant differences in streamflow. Despite these limitations, the approach conclusively demonstrates that hydrological models exhibit strong differences in their ability at generating streamflow on analogue catchments, and that the continental LSTM appears to best-fitted for climate change impact study, at least for the simple +6 °C and +20% scenarios.

## 5 Conclusion

This study compared four traditional hydrological models against two LSTM-based models across a domain of 148 catchments, focusing on projecting future streamflow. Over the past 25 years, traditional hydrological models have provided insights into the climate sensitivity of hydrological processes. However, results presented in this work suggest that LSTM neural networks emerge as a very effective tool for future climate change impact studies. Their ability to learn from and adapt to a wide range of climatic and hydrological conditions positions them as a better alternative to the hydrological models used in this study for accurately projecting future water resource conditions. Transitioning to LSTM models, however, necessitates careful consideration of their data needs, computational complexity, and the interpretability of their projections, highlighting the importance of continued research and methodological improvements in hydrological modeling in the context of climate change. Nevertheless, the era of traditional hydrological models in climate change studies is far from over. Although LSTM-based models seem better suited to the task, the future streamflow projections by both model types were similar in many instances, and the quantitative advantages of the LSTM models over the best performing traditional model were often minimal. Further research is essential to fully explore the potential benefits of LSTM-based models, particularly for catchments in different climate zones and the effects on streamflow extremes, which are crucial for many adaptation strategies.



**Code and data availability**

The hydrometeorological data for this study were sourced from the HYSETS database (Arsenault et al., 2020) https://doi.org/10.17605/OSF.IO/RPC3W. CMIP6 GCM model outputs can be obtained from the Earth System Grid Federation (ESGF) portal at Lawrence Livermore National Laboratory: https://esgf-node.llnl.gov/search/cmip6/. Processed data and the codes used in this research are available at https://osf.io/5yw4u.

**Author contribution**

JLM, FB, RA, and RT designed the experiments and JLM, FB, MCG and WA performed them. JLM, FB, and RA analyzed and interpreted the results with significant contributions from RT, EM, JPD, GRG, and LPC. JLM wrote the paper with significant contributions from FB and RA. RT, EM, JPD, GRG, and LPC provided editorial comments on initial drafts of the paper.

**Competing interests**

The authors declare that they have no conflict of interest.

**Acknowledgements**

The authors would like to thank the teams at the Direction Principale de l'Expertise Hydrique du Québec (DPEH) and Ouranos that made this project possible in the context of the INFO-Crue research program (project #711500).

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
