# Peer review of "Assessing the adequacy of traditional hydrological models for climate change impact studies: A case for long-short-term memory (LSTM) neural networks"

_EGUsphere, 2024_

## Author Comment (AC1)

**Assessing the adequacy of traditional hydrological models for climate change impact studies: A case for long-short-term memory (LSTM) neural networks**

We would like to thank the reviewers for their valuable and constructive feedback. We appreciate the time and effort that was put into the review. All major and minor concerns have been carefully addressed. Detailed responses to each of the reviewers' comments are presented below. For clarity, the reviewers' comments are presented in black font, with our responses in blue.

Sincerely,

Jean-Luc Martel, on behalf of all authors.

**Major modification: important note on significant changes, not affecting the main conclusions of the paper.**

The different reviewers highlighted a possible problematic aspect of the LSTM model implementation for this study, more specifically relating to the use of climate-based static descriptors in the model training and simulation. It was pointed out that using static climate-based descriptors in climate change was far from ideal and that they should be removed, or at least made to change along with the climate data. The same reasoning was proposed for the latitude and longitude static descriptors, where the model could learn some specificity for a given region rather than rely on the climate data to modulate flows. While we are expected to provide a detailed explanation on how we will revise the paper, this comment was too important, with the potential to influence the conclusion of our paper and needed to be investigated right away.

We therefore developed and trained a new model that does not use these data (climate-based statics and latitude/longitude descriptors). The only descriptors remaining are actual catchment descriptors such as drainage area, aspect, slope, elevation, soil porosity, land use and other such variables. The model was trained twice: for the regional LSTM model with only the initial 148 catchments (LSTM-R in the paper) and for the continental LSTM model using an extra set of 1000 catchments (LSTM-C in the paper). While we did not have the time to rerun all results, we tested the modified continental LSTM model (which incorporates additional donor catchments) on all 148 study catchments. This was conducted to assess performance over the reference period, using the four climate sensitivity experiments (+3 °C and +6 °C, -20 % and +20 % precipitation), as well as under the 22 climate models for the 2070-2099 period. The regional LSTM model (excluding additional donor catchments) was still running at the time of this writing. Additional optimization runs will also be completed before final publication. However, our results thus far clearly indicate that our conclusions remain unaffected by this methodological change. In the revised version, we will retain only the new runs (excluding latitude, longitude, and static climate variables), as we agree with the reviewers that this is a more reasonable approach for climate change impact studies. The various sections will be reworked accordingly.

[Figure]

**Figure RR1**: This figure is the same as Figure 3a in the original paper (model performance over the testing period), with the exception of the seventh boxplot, which corresponds to the modified continental model (with donor catchments (WD), excluding latitude and longitude (NLL), and no static climate descriptors (NS)). As shown, there are some minor differences compared to the latitude-included boxplots, but these are minor and align with the expected variance from multiple runs. We will also retrain the model to further optimize it. We anticipated some performance decline over the reference period due to the omission of these static descriptors and were somewhat surprised that it was not the case. This indicates that the essential information is contained within the hydrometeorological historical time series (P, T, Q) and the true static catchment descriptors, and the LSTM model can effectively capture this information without the excluded descriptors. This in itself is a very interesting result.

[Figure]

**Figure RR2**: This figure is the same as Figure 6 in the original paper (projected mean fall SON flows), with the exception of the seventh boxplot, which corresponds to the modified continental model (with donor catchments (WD), excluding latitude and longitude (NLL), and no static climate descriptors (NS)). This Figure shows that the modified LSTM keeps the same sensitivity to precipitation and temperature change. In particular, it preserves its added sensitivity to temperature compared to the process-based hydrological models.

[Figure]

**Figure RR3**: This figure is the same as Figure 7 in the original paper with the exception of the seventh boxplot, which corresponds to the modified continental model (with donor catchments (WD), excluding latitude and longitude (NLL), and no static climate descriptors (NS)). This figure contains the aggregated results from the 22 climate models. It shows that the modified LSTM aligns itself with the original two, thus preserving the conclusions drawn in the paper.

After careful analysis, we came to the following conclusion: The addition of static variables seems to help the model to converge faster in terms of training, but the actual impacts on hydrology are caused by the meteorological forcings. The static catchment attributes (area, slope, land-use, porosity, etc.) actually condition these flows and impact their regime. The climate static variables, on the other hand, do not impact the flow modulation and only seem to help the model understand the meteorological properties better in order to converge faster. By removing them, we not only made the model more robust (fewer inputs that change over time) but also made it harder and longer to train. However, in the context of climate change studies, we believe this version of the model makes more sense, and as such, we will modify all figures and results to include this version only. The revised paper will present "clean" figures, as these were made quickly for the revision.

**Reviewer #1:** https://doi.org/10.5194/egusphere-2024-2133-RC1

The study is sound and methodology is robust. The experimental design is clever, dataset curation (especially training/validation data setup) is suitable to answer the research questions and the rationality in choosing climate change scenarios and climate models is also reasonable. Language, presentation quality and significance are good, and results/conclusion are also sound and relevant. I recommend publication, however, after minor revisions.

Thank you very much for your positive comments and suggestions. Please refer to the point-by-point responses to your comments below.

1) you can merely infer from "between the lines" in the method section, that for the hydrological model runs evaluating climate change, the hydro-models are actually also trained on the hindcasts from the same GCMs that are used for the projections (as it should be done, so the model is trained from the same population that it is tested on). If it was done like this, then maybe just add a small paragraph or sentence to e.g. 2.4.2.3 to make this crystal clear. If I am mistaking and it is not done like this, we have a bigger problem and you need to redo the results and train on the hindcasts.

The training (or calibration) for the traditional hydrological models was performed on all available observations from the ERA5 reanalysis (i.e., precipitation and temperature between 1981 and 2007) using observed streamflow for the same period as the target variable. A small validation period between 2008 and 2012 was kept to have a fair comparison with the testing. Since hydrological models cannot be calibrated using climate model data (as there is no correlation between climate model data and observations on a day-to-day scale), the models are calibrated on the ERA5 reanalysis dataset (reanalysis of meteorological datasets, i.e. observations). Then, the climate model data is bias-corrected using the ERA5 data as reference, making it possible to run the model on the reference period GCM data. Finally, the bias-correction postprocessor is applied to the future climate model data and the resulting data is run in the hydrological model. This is a typical hydro climate processing chain, as used in the IPCC reports and most hydroclimatological sciences papers. Thus, it is not possible to calibrate directly on the climate model data directly, but the bias-correction step allows us to simulate the flow from the climate model in reference and future periods. This is also why we cannot evaluate day-to-day flows, but only long-term statistics, no matter the model (traditional or LSTM-based) used. This will be made clearer in the text.

2) The model seems overly complicated, given the fact that up to recently, the state of the art model only comprised one single LSTM layer (https://doi.org/10.5194/hess-25-2685-2021). It would be useful to benchmark (i.e. plot/compare) against the above-mentioned studies' performance to see if it actually gives an advantage in performance. The structure obviously doesn't hurt performance, as can be seen from the high scores shown in the manuscript, but does it help? As a comment without need for action: probably 90% of your network is inactive, but the remaining 10% is why it still works well.

It is true that a simpler model can give very good results as well, but trial and error have shown that adding complexity added more and more performance, although towards the end it was more marginal and thus, we stopped at this level. Also, the sheer amount of data points for training encourages the use of a larger model. We also implemented this architecture to act a bit like a

ResNet, using interconnected layers with residuals for a few layers, which helped robustness. The inconvenience is that it does add training time and demands more computing resources. To provide more concrete context, we did another run with a model similar to that of Kratzert et al. (2019), using a single LSTM layer with 256 cells, concatenated with statics and fed to a dense layer (rather than linear). We used the same 0.4 dropout and 0.001 learning rate. We also changed the 270-day look back window to a 365 day one to better account for the snow processes in Canada. The results show that the median testing KGE over all catchments was 0.71 (vs 0.77 for our model with no-donor catchments, and 0.82 with donor catchments). During the development of the models in the paper, we did this process with increasingly complex models and ended up with those used in the paper. We built upon this experience through various tests to provide this present model, even though it is indeed quite large and complex. But this is still a small model compared to models such as AlexNet and LLMs, and we believe that the addition of more data will always require more complex models to make use of it as best as possible. The figure below shows raw data for the various runs as in those in the "major modification" section at the top of this document, with the simple model runs in the last column for comparison.

[Figure]

**Figure RR4**: This figure is the same as Figure 3a in the original paper, with the exception of the seventh boxplot, which corresponds to the modified continental model (with donor catchments (WD), excluding latitude and longitude (NLL), and no static climate descriptors (NS)), and the eighth boxplot adding the simple LSTM model described above. This model was only trained once and it is quite probable that performing multiple trainings would improve results to a certain extent, but not to the point of competing with the larger models.

3) Chapter 4.4.1.: results and literature stand next to each other disconnectedly; no true conclusion is drawn in the chapter. Insert it.

Thank you for this suggestion. Based on the recommendations from all reviewers, the manuscript will be restructured. This section will be revised, and a clear conclusion will be provided in the revised manuscript.

4) Results and discussion are formally separated from each other, but then there is another bunch of analysis and four (!) figures in the discussion section, which renders the separation of results and discussion irrational. Either clearly separate results from discussion, or – much preferred, because much easier to follow/ understand in general – make a single "results and discussion" section and discuss noteworthy point right next to the figures.

Agreed. In the revised manuscript, we will combine the results and discussion sections into a single section, ensuring clear discussions aligned with related literature.

5) the conclusion is a stump and contains mostly commonplace statements. Revise to revolve around actual key conclusions from your results.

We will rework the entire conclusion section to better reflect the key conclusions that are presented throughout the different results.

6) typos:

Line 576 "mpdels" à models

The typo "mpdels" will be corrected to "models".

Line 671 "streamflow electricity" à elasticity

The typo "electricity" will be corrected to "elasticity", and it will be revised throughout the manuscript. The auto-correct was quite original on this one.

---

## Author Comment (AC2)

**Assessing the adequacy of traditional hydrological models for climate change impact studies: A case for long-short-term memory (LSTM) neural networks**

We would like to thank the reviewers for their valuable and constructive feedback. We appreciate the time and effort that was put into the review. All major and minor concerns have been carefully addressed. Detailed responses to each of the reviewers' comments are presented below. For clarity, the reviewers' comments are presented in black font, with our responses in blue.

Sincerely,

Jean-Luc Martel, on behalf of all authors.

**Major modification: important note on significant changes, not affecting the main conclusions of the paper.**

The different reviewers highlighted a possible problematic aspect of the LSTM model implementation for this study, more specifically relating to the use of climate-based static descriptors in the model training and simulation. It was pointed out that using static climate-based descriptors in climate change was far from ideal and that they should be removed, or at least made to change along with the climate data. The same reasoning was proposed for the latitude and longitude static descriptors, where the model could learn some specificity for a given region rather than rely on the climate data to modulate flows. While we are expected to provide a detailed explanation on how we will revise the paper, this comment was too important, with the potential to influence the conclusion of our paper and needed to be investigated right away.

We therefore developed and trained a new model that does not use these data (climate-based statics and latitude/longitude descriptors). The only descriptors remaining are actual catchment descriptors such as drainage area, aspect, slope, elevation, soil porosity, land use and other such variables. The model was trained twice: for the regional LSTM model with only the initial 148 catchments (LSTM-R in the paper) and for the continental LSTM model using an extra set of 1000 catchments (LSTM-C in the paper). While we did not have the time to rerun all results, we tested the modified continental LSTM model (which incorporates additional donor catchments) on all 148 study catchments. This was conducted to assess performance over the reference period, using the four climate sensitivity experiments (+3 °C and +6 °C, -20 % and +20 % precipitation), as well as under the 22 climate models for the 2070-2099 period. The regional LSTM model (excluding additional donor catchments) was still running at the time of this writing. Additional optimization runs will also be completed before final publication. However, our results thus far clearly indicate that our conclusions remain unaffected by this methodological change. In the revised version, we will retain only the new runs (excluding latitude, longitude, and static climate variables), as we agree with the reviewers that this is a more reasonable approach for climate change impact studies. The various sections will be reworked accordingly.

[Figure]

**Figure RR1**: This figure is the same as Figure 3a in the original paper (model performance over the testing period), with the exception of the seventh boxplot, which corresponds to the modified continental model (with donor catchments (WD), excluding latitude and longitude (NLL), and no static climate descriptors (NS)). As shown, there are some minor differences compared to the latitude-included boxplots, but these are minor and align with the expected variance from multiple runs. We will also retrain the model to further optimize it. We anticipated some performance decline over the reference period due to the omission of these static descriptors and were somewhat surprised that it was not the case. This indicates that the essential information is contained within the hydrometeorological historical time series (P, T, Q) and the true static catchment descriptors, and the LSTM model can effectively capture this information without the excluded descriptors. This in itself is a very interesting result.

[Figure]

**Figure RR2**: This figure is the same as Figure 6 in the original paper (projected mean fall SON flows), with the exception of the seventh boxplot, which corresponds to the modified continental model (with donor catchments (WD), excluding latitude and longitude (NLL), and no static climate descriptors (NS)). This Figure shows that the modified LSTM keeps the same sensitivity to precipitation and temperature change. In particular, it preserves its added sensitivity to temperature compared to the process-based hydrological models.

[Figure]

**Figure RR3**: This figure is the same as Figure 7 in the original paper with the exception of the seventh boxplot, which corresponds to the modified continental model (with donor catchments (WD), excluding latitude and longitude (NLL), and no static climate descriptors (NS)). This figure contains the aggregated results from the 22 climate models. It shows that the modified LSTM aligns itself with the original two, thus preserving the conclusions drawn in the paper.

After careful analysis, we came to the following conclusion: The addition of static variables seems to help the model to converge faster in terms of training, but the actual impacts on hydrology are caused by the meteorological forcings. The static catchment attributes (area, slope, land-use, porosity, etc.) actually condition these flows and impact their regime. The climate static variables, on the other hand, do not impact the flow modulation and only seem to help the model understand the meteorological properties better in order to converge faster. By removing them, we not only made the model more robust (fewer inputs that change over time) but also made it harder and longer to train. However, in the context of climate change studies, we believe this version of the model makes more sense, and as such, we will modify all figures and results to include this version only. The revised paper will present "clean" figures, as these were made quickly for the revision.

**Reviewer #3:** https://doi.org/10.5194/egusphere-2024-2133-RC3

The paper evaluates LSTM-based hydrological models and traditional hydrological models in Climate Change impact assessments. The models are assessed regarding their ability to simulate future streamflow and against their sensitivity to climatic changes. The authors conclude that LSTMs, given they are trained with sufficient data, are a viable and most likely a better alternative for climate change impact assessments.

Thank you very much for your comments and suggestions. We provide point-by-point replies to each of your comments below.

In my opinion the title does not fully express what you actually did in the study. You assessed the adequacy of both traditional models and LSTMs. Not sure, you might want to highlight that.

We agree that we assessed both lumped traditional hydrological models and LSTMs through the proposed methodology. However, our main goal was to highlight the potential effectiveness of LSTM neural networks compared to traditional hydrological models. Therefore, we consider that the current title reflects the intended take-home message for the readers.

The authors address an important topic and use a suitable dataset and methods for the evaluation. However, the main problem I have with the paper is the structure:

Nowhere in the paper do the authors refer back to the three objectives outlined at the end of the introduction. I would expect that you explain how you are going to achieve the objectives in the methods, show the respective results and discuss those, and ideally come back to the objectives in the conclusion. The paper seems unstructured in that regard and it is hard to understand which of the subchapters contributes to which objective, thus disconnecting the analysis from the objectives.

This is a valuable point also raised by Reviewers 1 and 2. Therefore, following the reviewers' comments, we will restructure the paper by combining results and discussion sections together, streamline the paper and move some figures in the supplemental materials.

Another structural problem is already evident in the abstract: Your last sentence of the abstract highlights the analysis of precipitation elasticity and catchment analogues. I like these two analyses, but I find it strange that these are shown (including figures) in the discussion only. I do not see a reason why these analyses should not be structured into methods, results, discussion.

This is a good point, and we will address it by combining the results and discussion sections, as mentioned in the previous reply.

Additional Comments:

l.130-134: It seems unclear at this point if the current study is also limited to >500km² and 30yr data. Suggest to add a brief explanation, particularly as you mention later (l.159) that you restrict to 20yr streamflow data.

This will be clarified by modifying this paragraph as follows: "*In the Arsenault et al. (2023) study, only those catchments with a drainage area exceeding 500 km$^2$ were included, thereby sidestepping potential issues related to scale and time lag in model regionalization efforts. Catchments also required at least 30 years of data over the 1979-2018 period to be selected in order to have sufficient data to train both the conceptual hydrological models and the deep learning implementations. These criteria were also used in this study, resulting in the selection of the same 148 catchments for the analysis.*". The 20-year limit was set only for the extra 1000 donor catchments to widen the set of available catchments for this analysis, but these were not as critical as the original 148 as they were not used for model testing. Therefore, this constraint was relaxed to 20 years for the extra set of catchments. This will be clarified in the text, especially at line 159.

l.136: I think the term 'scenario' is somewhat misleading in this context. I suggest to write something along the lines "An extra set of 1000 donor catchments was selected for an additional LSTM application."

Thank you for your suggestion. The term "*scenario*" will be replaced with "*LSTM configuration*" as suggested in the revised manuscript.

l.149: against the background of the previous sentence, it is unclear what you mean by 'common denominator for all models'

We will clarify the author's meaning by adding, "*(i.e., the one that corresponds to the intersection between both datasets)*" in the revised manuscript.

l.153-154: suggest to check Tarek et al. if that is generalizable for any catchment / region.

Thank you for your suggestion. The study presented by Tarek et al. was done for all North America, covering our entire study region. The study shows indeed a generally good performance over the whole region compared to observations, with some loss in performance in eastern U.S. compared to observations. Nonetheless, the performance remains satisfactory for the present study.

l.195: I assume the climatic descriptors are kept constant under climate change? Do you think considering a possible spatial shift of these climatic conditions under climate change could further improve the models? Perhaps a point worth discussing.

Please see the major modification presented at the beginning of this document for more information on how this was resolved and will be implemented in the revised version of the paper.

l.255: μ is missing in explanation

We will add in the revised manuscript that μ refers to the average of the simulation and observed streamflow, respectively.

l.289-290. I do not understand this part. Why is data combined from multiple catchments for the computation of the objective function? And why was the NSE used for the LSTM's objective function and not the KGE as for the classical hydrological models?

Since the model is a regional one (or continental for the one with the extra donors), it requires being fed data from various catchments. The order that these data are fed into the model for training is randomized to ensure stochasticity and to help convergence. Therefore, each batch needs to compute an objective function on a subset of the data sampled at random from hundreds of thousands of data points. To make this possible, the objective function needs to allow this. The objective function is a NSE-based metric that is slightly altered to ensure that the flow values are scaled appropriately, or else the larger catchments would weigh more than the smaller ones simply due to their size (and thus larger flows). Doing so with KGE is more complicated because the ratio of variability is highly impacted by the very small sample sizes of the batches. However, for evaluation, using KGE makes more sense as all flows are available, and the metric allows for more in-depth evaluation. For the traditional hydrological models, we would not use the same NSE-derived and scaled objective function as it is done one catchment at a time. Thus, we simply decided to use KGE directly. This means that the classical hydrological models are given a bit of an advantage over the LSTM models, but this is minor and, as the results show, the models are still comparable over the independent validation/testing period.

We propose to add a clarification to this effect in the paper.

l.305: Suggest to introduce LSTM-R and LSTM-C earlier in the manuscript and mention the respective 148 and 1000 catchments.

We will introduce LSTM-R and LSTM-C and the respective 148 and 1000 catchments configurations earlier in the manuscript as suggested.

l.322-327: I assume when implementing one test, all other variables were held constant (so no combination of TMP and PCP changes)? I suggest to mention that.

Yes, exactly. We will clearly state in the revised manuscript that no combination of temperature and precipitation changes was used in any of the tests. Changing both variables at the same time complicates the interpretation. The 22 GCMs provide projections where both variables change.

l.384-385: why no low flow metric, such as the annual minimum streamflow? Also, I suggest to add the time periods for which these metrics are calculated (I assume the hindcast and the future climate?)

In this work, we have computed and analyzed 51 different streamflow metrics. These metrics cover mean flow values, distribution quantiles, and low- and high-flow extremes. We chose to focus on mean annual and seasonal streamflows, as these are robustly simulated by process-based hydrological models and serve as key climate change indicators. Low flows present challenges for process-based models, with a high level of uncertainty in projecting low flows in a warmer climate, where hydrological model uncertainty is much larger than that of GCMs. We also included the mean of maximum annual streamflow, which, as a "moderate extreme," is well represented by hydrological models. More extreme low- and high-flow metrics would require additional validation, especially for LSTM models. We will provide a stronger justification for our choice of metrics in the revised manuscript.

l.395-397: You mention NSE and KGE at three locations now. I am confused what data and models are used for which metric. And if different metrics were used for different models, I see a significant bias here given that you compare other metrics (see 2 comments further below). I suggest to mention the KGE and NSE metrics only once in the manuscript to avoid confusion. Also, in l.403 you mention NRMSE which was not mentioned in the methods.

Considering that the objective function used to train the LSTM model incorporates all catchments at the same time, it is necessary to add a standard deviation weighting to avoid overfitting catchments with larger streamflow. No identical objective function can be fully used for both types of models. In all cases, the traditional hydrological models are favored by this methodological choice, as mentioned previously.

We also propose to add the NRMSE (as well as the NSE which was not clearly mentioned) in the method section to avoid any confusion.

l.410: You did not introduce the optimum values for each metric. I suggest to add this information to where the metrics are introduced first, or/and the optimum could be added as a line to the diagrams.

We will add the optimum values for each of the metrics in section 3.1 of the revised manuscript as suggested.

l.431: Could the difference you see in the variability ratio between conventional and LSTM models be due to the different objective functions you used (KGE vs NSE)? The tradeoff between the different performance criteria is interesting. For further discussion of the relationship between the performance criteria, you can look into Guse et al. 2020 (https://doi.org/10.1080/02626667.2020.1734204)

This is a good idea, and we will analyze the impact of the objective function on the variance ratio for the various models. We will then add these findings in the revised manuscript to better contextualize these results.

l.442-443: suggest to summarize the main results of the other metrics here. I would assume the QMM is of interest to many readers.

This is a good idea, and we will summarize the results for readers.

l.465: I do not understand why a more pronounced response to temperature changes implies providing more accurate projections. Without comparing this to observations, I think this is not defendable. Please explain.

We agree with your comment. This was definitely worded incorrectly. We will revise this paragraph accordingly.

l.500-501: I am not sure about this statement. The classical hydrological models project an increase in streamflow. That would mean precipitation is overruling temperature.

This comment was mostly based on the relative position of all 6 models. We will rephrase and expand to also discuss the increase/decrease of streamflow as shown on the Y-axis.

l.509: what do you mean by "near the surface"?

"Near the surface" refers to a height of 2 meters above the surface. We will clarify this as "near the surface (2 meters height)" in the revised manuscript.

l.532: Why did you evaluate this section only for QMA?

This paper is already too long (as mentioned previously by the reviewers). We have decided to select this variable to conduct this analysis. However, all data are available in the data repository and readers could evaluate the metrics of their choosing to determine if the model is appropriate for their needs. Internally, we analyzed 51 metrics (The 51 from Table 2 in Arsenault et al. 2020) and results are fairly consistent. We decided to only show QMA for space considerations, given that it is one of the main metrics used to evaluate impacts of climate change on hydrology. Had we gotten strange and unexplainable results for QMA, it would have been a clear sign that the LSTMs were perhaps not suited for climate change studies. Doing more metrics would be desirable but we must balance the amount of data and results presented with the overall message, we wish to expose.

Arsenault, R., Brissette, F., Chen, J., Guo, Q. and Dallaire, G., 2020. NAC2H: The North American climate change and hydroclimatology data set. Water Resources Research, 56(8), p.e2020WR027097.

l.561: I do not understand what you mean with "to prevent contamination"

This refers to "data leaking", meaning that any data from the testing or validation sets cannot be used to help build the scalers or be used in the process of defining the model weights. Doing so would let the model include that data during training and thus make it more robust in testing, meaning that the model would be "cheating" by accessing "future" data it should know nothing about. This will be clarified in the paper, along with appropriate references.

l.576-577: Why do you write here that the LSTMs only use the climatic data? I suggest to add "..rain, and snow besides the catchment descriptors represent a fraction...".

Good point, we will add the descriptors to this list in the revised paper.

l.582-585: There are many hydrological models around that can make use of additional physical and temporal data. While I understand the advantage of complex LSTMs that can ingest all this data, I think you should not overstress this 'advantage' only because you chose traditional lumped hydrological models only, that cannot use additional data.

Actually, this section of the text points to a limitation of the LSTM models, in that while they can use a large variety of input data, that data also needs to be available for any simulation, forecast, or projection. This is a limiting factor that other models do not have (i.e. you could calibrate a hydrological model using remote sensing data to tweak parameters and then feed climate data to the model with no issue, but this is impossible with the LSTM-based models used here). Therefore, we propose to leave the text as is. As to the comment related to the ingestion of data, we will clarify elsewhere in the text (see one of the comments below) that other hydrological models can use more sources of hydrometeorological data than those used in this study.

l.625-629: Isn't it also reassuring that the different classical hydrological models performed similar? It could also mean that the projections can be considered robust.

This is a good point, and will be added to the final text.

l.634-635: Well, this is clearly beyond scope for your study, but you could at least discuss that there are studies that used historical streamflow change to evaluate models (see for instance Eyring et al. 2019, https://doi.org/10.1038/s41558-018-0355-yor; Kiesel et al 2020, https://doi.org/10.1007/s10584-020-02854-8 who both propose out-of-sample evaluations). And also, Krysanova et al (2018, already cited in your manuscript) provide a 5-step validation procedure that allows an assessment of how well models are suitable for climate change impact assessments. This might also be worth discussing.

This is a good idea in theory, but the problem is that we need to find a large set of catchments that have similar climate change signatures, and thus require long enough time periods. We can add a sentence in the paper to that effect.

l.646-650: I do not completely agree with this statement based on your study, considering that other traditional models exist that utilize additional data sources as well. I suggest to add to l.648: ... a theoretical advantage over the four traditional hydrological models used in this study.

We agree to a certain extent, but we still believe the theoretical advantage holds. The advantage comes not from the fact that more data variables can be ingested (which other models can do, such as humidity, radiation, wind and other such variables). The advantage comes from the fact that the LSTM model ingests data from over 1000 diverse catchments and can learn how different climates lead to different streamflow values. No other model can do this, unless we count large-scale regional models that are calibrated on a large and diverse region. But even then, the parameters are fixed in time and the changed meteorological data is still fed to a deterministic model that cannot modulate flows according to nonlinear processes between meteorology and hydrology. This is where the LSTM strives. We therefore propose to keep this in the text but to better contextualize it, as done in this response.

l.660: Why is this a disadvantage of conceptual hydrological models? You also fixed the LSTM parameters after training?

Yes, but it is adaptive and has many internal weights able to process the various seasonalities within. This would be more similar to a hydrological model whose parameters are calibrated to different values for every day, for example. This will be clarified in the text.

l.685-694: in l.690, could you mention percentages instead of "a few" or "most"? Also, you could simply calculate the elasticity ratios yourself based on your the historical data?

We can calculate the percentages indeed, and will do so in the revised version. However, calculating the elasticity is a difficult and time-consuming prospect given the detailed analysis required to determine the impact of a change of precipitation on the change in streamflow when many confounding variables are at play (antecedent soil moisture, snow, vegetation, etc.). We will therefore only rely on the existing literature on these values.

l.773-777: I suggest to also mention that you did not include more physically-based hydrological models that can utilize additional data which could react differently to the future climate forcings.

We will mention that more physically-based hydrological models were not included in this study due to the extensive work and computing power required to calibrate such models on 148 catchments. Nonetheless, we will highlight that different results can be obtained, as more physically-based hydrological models can utilize and process additional data such as elevation, land types, land uses, and more.

Minor comments:

l.31 consider "studies evaluate" to avoid repeating "assess"

We will replace "studies evaluate" to "assess" as suggested in the revised manuscript.

l.182 comprehnsive -> comprehensive

We will correct this typo in the revised manuscript.

l.280 ...performance gains in other... ?

We will correct this typo in the revised manuscript

l.370 you express pcp change in %, therefore *100 should be added to the calculation example.

We will add " *100 " as suggested.

l.511: wet vs dry models

We will correct this typo in the revised manuscript

l.576: mpdels -> models

We will correct this typo in the revised manuscript

l.671: electricity -> elasticity

We will correct this typo in the revised manuscript

l.720: unit should be m3 s-1 km-2 ?

This is correct; we will correct this typo in the revised manuscript.

l.730 and supplementary material Figures: suggest to add or mention the RMSE unit of normalized streamflow.

We will add this clarification in the revised manuscript.

---

## Author Comment (AC3)

**Assessing the adequacy of traditional hydrological models for climate change impact studies: A case for long-short-term memory (LSTM) neural networks**

We would like to thank the reviewers for their valuable and constructive feedback. We appreciate the time and effort that was put into the review. All major and minor concerns have been carefully addressed. Detailed responses to each of the reviewers' comments are presented below. For clarity, the reviewers' comments are presented in black font, with our responses in blue.

Sincerely,

Jean-Luc Martel, on behalf of all authors.

**Major modification: important note on significant changes, not affecting the main conclusions of the paper.**

The different reviewers highlighted a possible problematic aspect of the LSTM model implementation for this study, more specifically relating to the use of climate-based static descriptors in the model training and simulation. It was pointed out that using static climate-based descriptors in climate change was far from ideal and that they should be removed, or at least made to change along with the climate data. The same reasoning was proposed for the latitude and longitude static descriptors, where the model could learn some specificity for a given region rather than rely on the climate data to modulate flows. While we are expected to provide a detailed explanation on how we will revise the paper, this comment was too important, with the potential to influence the conclusion of our paper and needed to be investigated right away.

We therefore developed and trained a new model that does not use these data (climate-based statics and latitude/longitude descriptors). The only descriptors remaining are actual catchment descriptors such as drainage area, aspect, slope, elevation, soil porosity, land use and other such variables. The model was trained twice: for the regional LSTM model with only the initial 148 catchments (LSTM-R in the paper) and for the continental LSTM model using an extra set of 1000 catchments (LSTM-C in the paper). While we did not have the time to rerun all results, we tested the modified continental LSTM model (which incorporates additional donor catchments) on all 148 study catchments. This was conducted to assess performance over the reference period, using the four climate sensitivity experiments (+3 °C and +6 °C, -20 % and +20 % precipitation), as well as under the 22 climate models for the 2070-2099 period. The regional LSTM model (excluding additional donor catchments) was still running at the time of this writing. Additional optimization runs will also be completed before final publication. However, our results thus far clearly indicate that our conclusions remain unaffected by this methodological change. In the revised version, we will retain only the new runs (excluding latitude, longitude, and static climate variables), as we agree with the reviewers that this is a more reasonable approach for climate change impact studies. The various sections will be reworked accordingly.

[Figure]

**Figure RR1**: This figure is the same as Figure 3a in the original paper (model performance over the testing period), with the exception of the seventh boxplot, which corresponds to the modified continental model (with donor catchments (WD), excluding latitude and longitude (NLL), and no static climate descriptors (NS)). As shown, there are some minor differences compared to the latitude-included boxplots, but these are minor and align with the expected variance from multiple runs. We will also retrain the model to further optimize it. We anticipated some performance decline over the reference period due to the omission of these static descriptors and were somewhat surprised that it was not the case. This indicates that the essential information is contained within the hydrometeorological historical time series (P, T, Q) and the true static catchment descriptors, and the LSTM model can effectively capture this information without the excluded descriptors. This in itself is a very interesting result.

[Figure]

**Figure RR2**: This figure is the same as Figure 6 in the original paper (projected mean fall SON flows), with the exception of the seventh boxplot, which corresponds to the modified continental model (with donor catchments (WD), excluding latitude and longitude (NLL), and no static climate descriptors (NS)). This Figure shows that the modified LSTM keeps the same sensitivity to precipitation and temperature change. In particular, it preserves its added sensitivity to temperature compared to the process-based hydrological models.

[Figure]

**Figure RR3**: This figure is the same as Figure 7 in the original paper with the exception of the seventh boxplot, which corresponds to the modified continental model (with donor catchments (WD), excluding latitude and longitude (NLL), and no static climate descriptors (NS)). This figure contains the aggregated results from the 22 climate models. It shows that the modified LSTM aligns itself with the original two, thus preserving the conclusions drawn in the paper.

After careful analysis, we came to the following conclusion: The addition of static variables seems to help the model to converge faster in terms of training, but the actual impacts on hydrology are caused by the meteorological forcings. The static catchment attributes (area, slope, land-use, porosity, etc.) actually condition these flows and impact their regime. The climate static variables, on the other hand, do not impact the flow modulation and only seem to help the model understand the meteorological properties better in order to converge faster. By removing them, we not only made the model more robust (fewer inputs that change over time) but also made it harder and longer to train. However, in the context of climate change studies, we believe this version of the model makes more sense, and as such, we will modify all figures and results to include this version only. The revised paper will present "clean" figures, as these were made quickly for the revision.

**Reviewer #2:** https://doi.org/10.5194/egusphere-2024-2133-RC2

The study aimed to compare process-based models (PB) and machine learning models (ML) when used for scenario analysis. Specifically, they assessed three common PBs and two Long-Short Term Memory (LSTM) configurations. The researchers focused on examining the sensitivity of streamflow to different forcing perturbations. Their findings indicated that LSTM trained at a continental scale is a more reliable model due to its training on a wider range of variability. However, the study suggested that in most cases, the sensitivity between PB and ML is similar, with a few exceptions.

Thank you very much for your comments and suggestions to improve the manuscript. Below, we provide point-by-point responses to all concerns, comments and suggestions.

Major comments:

1. The use of latitude, longitude, and climatic static attributes in the LSTM models restricts the model to learn local information rather than capturing the spatial variability in the dataset. This could affect the model's ability to accurately analyze sensitivity under different climatic conditions. I suggest running the model without latitude and longitude and using the minimum number of climatic attributes while changing them according to the sensitivity being analyzed (ex. 20% increase in precipitation means a 20% of the mean annual precipitation).

This is a good observation. We were curious about this when reading your comments and have retrained the model without using the latitude and longitude in the variables. Please see the "major modifications" section at the top of this document for an in-depth response to this point and how the paper will be modified to reflect this.

2. There is an excessive number of figures and sections presented. Some results are repetitive, and certain sections may not add significant information due to high uncertainty (ex. 4.4.2). I recommend including only the figures and sections that directly support the main conclusion of the report while considering moving additional figures and analyses to the appendix or supplemental information.

Based on comments from Reviewers 1 and 3, we will rework and combine the results and discussion sections and move some of the figures into the supplemental information.

Minor comments:

Line 14. The comment about traditional hydrological models relying on historical climate data is misleading. ML models rely on historical data too.

Thank you for pointing this out. The sentence will be reworded in the revised manuscript as follows: "*Traditional hydrological models, which rely on simplified process parameterizations with a limited number of parameters, are scrutinized for their capability to accurately predict future hydrological streamflow in scenarios of significant warming.*"

Line 85. The definition of what is long and short is subjective. In fact, how long the memory in an LSTM model remains an open question.

*This is a good point. The sentence will be reworded to "Their unique architecture enables them to learn and remember over longer sequences of data compared to standard RNNs, making them highly effective for predictions of time series" in the revised manuscript.*

Line 109. The problem is not the new climatic scenario, the problem is how to define when LSTM is in extrapolation mode. Because we can have an extreme condition in one catchment but the same would be normal in another, so the model can infer the relationship. In that case, the model is still in interpolation mode.

*Agreed. We will reformulate this in the revised paper. The advantage of using all of these extra donor catchments is to widen the training dataset characteristics such that the climate model-driven simulations are mostly in interpolation mode rather than extrapolation.*

Line 118. You should have an introduction between sections and subsections. Probably a title as a dataset or data would match better with what you have.

*We will add a short introduction between sections and subsections. However, we do not understand the second part of your comment. The section is titled "Study area and Data", as we describe the study area in 2.1 and the data in 2.2.*

Line 132. More data is better, but how did you define this number?

*It was not possible to take more than approximately 30 years of data without having to drop a significant number of watersheds. We will mention this in the revised manuscript as follows: "Catchments also required at least 30 years of data over the 1979-2018 period to be selected in order to have sufficient data to train both the conceptual hydrological models and the deep-learning implementations. The basin selection criterion was set to a minimum of 30 years of data to ensure not only a sufficient data length but also a robust sample of basins for performance assessment."*

Line 161 – 166. You do not need that paragraph. You can use a reference to explain this in more detail.

*We will make the paragraph significantly shorter and add the following reference: Wickham, H., & Stryjewski, L. (2011). 40 years of boxplots. Am. Statistician, 2011.*

Figure 1. It needs a legend.

*We will add a legend explaining the difference between study catchments and donor catchments (large circles with black outline vs small circles with gray outlines for the a and b panels, and green and orange boxes for the c and d panels respectively).*

Line 193. This attribute shouldn't be used because you are anchoring the dynamic to a location which is exactly what you are trying to avoid. That can have serious effects on the sensitivity of your model. AND Line 195. How are you disentangling the correlation between catchment attributes and the meteorological forcing?

These two comments are related to the major comment related to the latitude and longitude statics as well as the climate-based static descriptors. Please see the major comment at the beginning of this document for a detailed response to this point.

Line 317. How did you apply that modification? only testing or in the entire period?

This was done over the entire period. This will be specified in the revised manuscript.

Line 404. This is not the reason for not presenting the validation period. All your results should be in a period that was never used during the training and validation.

This is correct. We proposed the following reformulation for this sentence: "Note that the validation period for the LSTM-based models is not shown as the validation data are contaminated by the training data, and thus, should not be investigated."

Figure 3. I recommend using CDF plots. This format has been widely used in streamflow models. I suggest adding a line where the best value is found.

We will test using CDFs, and if the graphs are still clear enough to interpret (i.e. not stacked too heavily one atop the other), we will consider using CDFs instead of boxplots.

Line 453 – 454. This could be a consequence of using latitude and longitude as input. The LSTM model is fixed to the location so it is less sensitive to precipitation because part of the precipitation correlation is shared with those attributes or with the climatic ones.

This is a good comment and will be resolved in the revised version. Please see the main major comment at the beginning of this response document for information on how this was handled.

Line 461 – 462. Something similar could be happening here with elevation. Temperature and elevation are very correlated. An interesting experiment would be increasing temperature and decreasing elevation by using an altitudinal gradient. Should the sensitivity increase or decrease?

See Supplemental material Figure S2 b) to see the distribution of elevation across all watersheds. All watersheds are within a 500-meter range more or less, thus the impact of elevation on temperature should be marginal, much less so than the impacts of climate change.

Figure 4. I would decrease the y-axis range. Probably [-60,20] for a and b. [-50,50] for c and d.

Agreed, the y-axis ranges in Figure 4 will be modified as suggested to improve the display of the results.

Figures 5 and 6. Move it to the appendix or supplement information.

We will modify the general structure of the manuscript based on all reviewers comments. Please refer to our reply to your main comment 2 for further details.

Line 479 – 480. Explain why it is clear to you.

This is based on Figure 2 which shows mean annual temperature and precipitation changes for each of the 22 GCM models. Median temperature increases range from +4 to +8.5 °C and from +4 % to +20 % for precipitation. Based on this, the +6 °C and +20 % sensitivity scenarios are a lot more realistic than the +3 °C and -20 % ones. We will rephrase and clarify this statement.

Figure 7. In many cases the differences look not significant, you should do hypothesis testing to check the level of significance. Moreover, you should mention something about the differences in the variability between some models. All the sub-figures must have the same y-axis range to do a fair comparison. I do not think you need all the sub-figures; you should show here only the most significant.

Good point for the figures. We will rearrange Figure 7 so that the range (maxY - minY) is the same for all 6 panels. We will also double-check the other similar Figures for the same issue. We performed statistical tests prior to submitting the preprint. More specifically, the nonparametric rank sum test for equality of median was used. The test showed that the LSTM results were in almost all cases statistically significantly different from the process-based models. We will expand on this in the revised version.

Line 537. No result can be considered a result, but in that case, you could move the entire section to supplemental information.

In this case, we prefer keeping this analysis here as it also supports the fact that the LSTM-based models and the classical hydrological models are providing similar responses to the same inputs. The fact that the results show no difference is actually very positive for the analysis. We therefore propose to leave this as-is, albeit in the new "results + discussion" section as discussed in another set of comments.

Line 558. But exactly for this reason you have the third period. Are you trying to say that this period is not representative enough? If this is the case, you should check that.

This sentence aims to convey the fact that in typical hydrological modelling, we have 2 phases: calibration and validation. But in Deep Learning, there are 3 phases required: Training (equivalent to calibration), validation, and testing (equivalent to validation for hydrological models). There is no equivalent of the "DL-Validation" in the hydrological modelling world. This data is used as a stopping criteria only, and is not used to define the scaler parameters nor is it used as training data to determine the model weights. It is used to stop the training process once the best validation is achieved, to prevent overfitting. Without this, models would converge to near perfection in training but would be completely useless in testing due to the massive overfitting. This will be clarified in the text.

Line 559 – 561. This is not part of the discussion; this is just part of the methodology.

We will remove this statement from the discussion and incorporate it into the LSTM model description in the revised manuscript.

Line 563 – 564. This is a strong statement without support. Remember that the number of parameters is not comparable between PB and ML models.

The vastly different number of parameters is what is referred to in the term "significantly greater number of degrees of freedom" in the original text. We proposed to rephrase the sentence as follows: "*From Figure S3, it is evident that LSTM training period performance surpasses that of conceptual models.*" and remove the notion of degrees of freedom.

Line 568 – 569. This sentence is exactly the opposite you said in line 558. You should put everything in one paragraph to tell a more consistent story.

Yes, the sentence is opposite to the previous one because the first one refers to the training period (i.e. calibration) where the performance can be essentially perfect with a large enough model. We state that the training performance should not be analyzed deeply because it is essentially unusable to verify model performance, contrarily to traditional hydrological models. In lines 568-569, we are talking about the independent testing performance, which can be used to assess model performance and where no signs of overfitting are seen. This is due to the precautions taken to prevent overfitting, using the validation period (the DL-validation, not the validation of the hydrological model) as a stopping criteria for training. We will rewrite this section to better clarify the intent.

Line 571 – 572. 1000 sounds like a large number of catchments so I disagree.

This sentence simply states that we saw an important improvement in model results when going from 148 to 1148 (+1000) catchments, and thus this confirms that adding a large number of catchments improves performance. We do not understand what this comment is referring to that could be disagreed with.

Line 575 – 576. Are you talking about distributed models? If this is the case, this would not be a fair comparison. If you want to add more degrees of freedom to PB, for example, you could combine the different sources of precipitation. Moreover, remember that the parameters in a PB encode the local descriptors (local characteristics) that the model needs.

It is true that we could use more sources of data to generate ensembles for example, which would allow averaging using a model averaging system that could indeed add more degrees of freedom. However here we meant that a single model simulation of a PB cannot make use of multiple precipitation datasets at once unless that model was specifically built to handle multiple precipitation inputs at once. The structure is fixed and cannot be increased. LSTM models afford this capability. And yes, the PB models encode the information directly into the model structure, but this also means that these are fixed in space and time, whereas an LSTM model can be applied

to an ungauged location by using the descriptors, and can add more if needed. Ex. some PBs do not use the soil porosity or hydraulic conductivity explicitly, whereas an LSTM model could add one with no issue as long as the data is available.

Line 580. I disagree. More variables increase performance but decrease interpretability. How could you do the same sensitivity analysis with 10 hydroclimatic variables highly correlated?

We agree with this, the interpretability would be much lower with many highly correlated variables. But this is not what we are advocating. We are expressing that adding mode data could (or most likely would, based on other papers) increase model performance, and could make the model more likely to respond accurately to climate change signals due to the extra information. A model using only precipitation would definitely see improvements in adding temperature. We are simply extending this reasoning to higher orders.

Line 590 – 591. You do not need to be sorry for finding that those models are the worst, this is just part of the results. Delete the sentence.

This sentence will be deleted.

Line 605 – 610. I disagree. The situation is exactly the opposite. You are not counting all the sensibilities; a temperature change can change the precipitation too, meaning that the final sensitivity of streamflow can be higher or lower. So, your analysis is a simplified sensitivity analysis which does not mean you are more accurate.

We certainly did not mean to imply that varying one-parameter at a time is better despite our poor choice of words. Of course, a temperature increase definitely affects precipitation in a myriad of different ways (mean, variability, extremes). A sensitivity analysis typically varies one parameter at a time, and we still believe that this is a useful approach (as stated in the paper) to better understand the potential difference in sensitivity between the classic hydrological models and the LSTM class of models.

We will rephrase the original sentence: "*By independently altering each variable—precipitation and temperature—we were able to more accurately evaluate the impact of each change…*"
with the following:

"*By independently altering each variable—precipitation and temperature—we were able to quantify the impact of each change, ...*"

Line 622. What family is that?

We will rephrase to: "*The four traditional hydrological models share similar structures, all being lumped conceptual models*".

Line 625. This is a contra argument about sensitivity coming from structure.

We are not entirely sure what this comment applies to. This sentence aims to show that the sensitivity was similar across the 6 models, including 4 conceptual ones with similar structures, and 2 LSTM-based models with extremely different structures. Thus, we do not think the sensitivity comes from the structure, but from the meteorological inputs.

Section 4.4. You focus just on which is the best. You must analyze the benefit of the ensemble of models (multi-representation approach). The concept of the best model does not exist.

While a multi-model approach can be used to provide a better evaluation of the overall uncertainty, our results support the conclusion that when using a single hydrological model, the LSTM-based model is likely to provide a better evaluation of the impact of climate change on the hydrological process. However, we agree that adding a section in the text highlighting the potential of a multi-model approach to evaluate the overall uncertainty is important. Also, we will propose to reformulate the section heading as follows: "Which type of hydrological models should we trust more for climate change impact studies?"

Line 653. That is not true. There is a lot of research on interpretability. We do not have yet the same level of interpretability as PB, but this does not mean we are not going to get it in the future.

This is a correct assessment. For some simple deep learning models there is starting to be some progress in interpretability, and the statement will be changed to clarify that this is not realistic at the moment but that with time this might be something that becomes possible.

Figure 9. I would prefer a table or a CDF figure showing the distribution. It is very hard to compare models within each group.

We will test using CDFs, and if the graphs are still clear enough to interpret (i.e. not stacked too heavily one atop the other), we will consider using CDFs instead of boxplots.

Line 701 – 703. I agree about the catchment attributes used however you considered only input similarity. This is not enough to define dynamic similarity, at least you should consider adding similarity in the streamflow signature.

We understand the reviewer comment. However, this is not possible for this assessment. The reason is that we are trying to see if the simulated flows under a future climate are well represented by the LSTM model. To do so, we need to find indicators that are independent of streamflow, as using a streamflow signature metric to find an analogue would ensure that the catchments found had similar hydrographs. This would defeat the purpose of ensuring that the analogues in terms of climate are indeed analogues in terms of hydrology.

Line 705. If this is the case, you should use a uniform weight distribution. Add information supporting your decision to use something different than uniform.

All of our catchments are located within Northeastern North America. A simple look at North American maps of mean annual precipitation and temperature clearly shows that mean annual precipitation changes little from North to South, whereas there is a huge mean annual temperature gradient therefore justifying our choice of weighing temperature more. As described in the paper, we tried different weighting schemes and picked the one which seems to result in the best analogues based on a limited number of catchments. Our evaluation was qualitative. Ultimately, the results were similar (with the continental LSTM model performing slightly better), even with a choice of analogues that did not seem optimal. We have to rerun the analogues with the newly developed LSTM models and will consider using equal weightings in the paper to make it simpler.

Figure 10. Given the huge difference between the analogues and the models, it is impossible to say that one model is better than the other. Moreover, if I suppose that the catchments presented are the best ones, it is impossible to infer more from this type of comparison.

Analogue analyses are, by definition, uncertain. We provide this analysis with the stated aim of finding adequate analogues (10) for each case to encompass the uncertainty of selecting the "best" analogue catchments. Looking at the hydrographs, it is clear that the models are all representing the hydrological cycle correctly, albeit with some key differences in select aspects of the hydrographs. We can definitely see that LSTM models perform better in some cases and classical models perform better in others, which is expected. However, it is clear that the LSTM based models are not consistently worse in any part of the hydrograph, but present a balanced comparison with the hydrological models. This is reassuring, in that the objective of the paper is to investigate if LSTM-based models can be used for climate change impact studies, and this analysis seems to confirm that they are at least not worse than classical hydrological models.

Line 759 – 764. Exactly for this reason I would drop the entire section. The results from this section are not different than before but with a huge uncertainty.

We respectfully disagree about dropping the entire section. While there are indeed some limitations to this analysis, it still provides valuable insight on the ability of the different types of hydrological models to simulate streamflow under a changing climate. It is one of the only insightful methods to assess the ability of models to represent climate change and to validate those results. Furthermore, these results are supported by the other results from this paper and thus contribute to the overall objectives of the paper. There is definitely some uncertainty, but this does not warrant, in our opinion, removing the analysis and its findings.

---

## Author Response (AR1)

**Assessing the adequacy of traditional hydrological models for climate change impact studies: A case for long-short-term memory (LSTM) neural networks**

We would like to thank the reviewers for their valuable and constructive feedback. We appreciate the time and effort that was put into the review. All major and minor concerns have been carefully addressed. Detailed responses to each of the reviewers' comments are presented below. For clarity, the reviewers' comments are presented in black font, with our responses in blue.

Sincerely,

Jean-Luc Martel, on behalf of all authors.

**Major modification: important note on significant changes, not affecting the main conclusions of the paper.**

The different reviewers highlighted a possible problematic aspect of the LSTM model implementation for this study, more specifically relating to the use of climate-based static descriptors in the model training and simulation. It was pointed out that using static climate-based descriptors in climate change was far from ideal and that they should be removed, or at least made to change along with the climate data. The same reasoning was proposed for the latitude and longitude static descriptors, where the model could learn some specificity for a given region rather than rely on the climate data to modulate flows. As was discussed in our replies to the initial reviewer comments, we opted to perform a quite major change to the methods and redo the work without those descriptors (latitude, longitude and climate-based indicators). The only descriptors remaining are actual physical catchment descriptors such as drainage area, aspect, slope, elevation, soil porosity, land use and other such variables.

The model was trained twice: for the regional LSTM model with only the initial 148 catchments (LSTM-R in the paper) and for the continental LSTM model using an extra set of 1000 catchments (LSTM-C in the paper). In the revised version, we retained only the new runs (excluding latitude, longitude, and static climate variables), as we agree with the reviewers that this is a more reasonable approach for climate change impact studies. The various sections have been reworked accordingly. Figures and results were therefore all changed, although conclusions remain the same. The track-changes version of the document shows the breadth of the changes, which we believe are now more robust and in-line with the stated objectives related to LSTM usage for modelling the impacts of climate change in hydrology.

**Reviewer #1:** https://doi.org/10.5194/egusphere-2024-2133-RC1

The study is sound and methodology is robust. The experimental design is clever, dataset curation (especially training/validation data setup) is suitable to answer the research questions and the rationality in choosing climate change scenarios and climate models is also reasonable. Language, presentation quality and significance are good, and results/conclusion are also sound and relevant. I recommend publication, however, after minor revisions.

*Thank you very much for your positive comments and suggestions. Please refer to the point-by-point responses to your comments below.*

1) you can merely infer from "between the lines" in the method section, that for the hydrological model runs evaluating climate change, the hydro-models are actually also trained on the hindcasts from the same GCMs that are used for the projections (as it should be done, so the model is trained from the same population that it is tested on). If it was done like this, then maybe just add a small paragraph or sentence to e.g. 2.4.2.3 to make this crystal clear. If I am mistaking and it is not done like this, we have a bigger problem and you need to redo the results and train on the hindcasts.

*The training (or calibration) for the traditional hydrological models was performed on all available observations from the ERA5 reanalysis (i.e., precipitation and temperature between 1981 and 2007) using observed streamflow for the same period as the target variable. A small validation period between 2008 and 2012 was kept to have a fair comparison with the testing. Since hydrological models cannot be calibrated using climate model data (as there is no correlation between climate model data and observations on a day-to-day scale), the models are calibrated on the ERA5 reanalysis dataset (reanalysis of meteorological datasets, i.e. observations). Then, the climate model data is bias-corrected using the ERA5 data as reference, making it possible to run the model on the reference period GCM data. Finally, the bias-correction postprocessor is applied to the future climate model data and the resulting data is run in the hydrological model. This is a typical hydro climate processing chain, as used in the IPCC reports and most hydroclimatological sciences papers. Thus, it is not possible to calibrate directly on the climate model data directly, but the bias-correction step allows us to simulate the flow from the climate model in reference and future periods. This is also why we cannot evaluate day-to-day flows, but only long-term statistics, no matter the model (traditional or LSTM-based) used. This has been made clearer in the text.*

2) The model seems overly complicated, given the fact that up to recently, the state of the art model only comprised one single LSTM layer (https://doi.org/10.5194/hess-25-2685-2021). It would be useful to benchmark (i.e. plot/compare) against the above-mentioned studies' performance to see if it actually gives an advantage in performance. The structure obviously doesn't hurt performance, as can be seen from the high scores shown in the manuscript, but does it help? As a comment without need for action: probably 90% of your network is inactive, but the remaining 10% is why it still works well.

*It is true that a simpler model can give very good results as well, but trial and error have shown that adding complexity added more and more performance, although towards the end it was more marginal and thus, we stopped at this level. Also, the sheer amount of data points for training encourages the use of a larger model. We also implemented this architecture to act a bit like a*

ResNet, using interconnected layers with residuals for a few layers, which helped robustness. The inconvenience is that it does add training time and demands more computing resources. To provide more concrete context, we did another run with a model similar to that of Kratzert et al. (2019), using a single LSTM layer with 256 cells, concatenated with statics and fed to a dense layer (rather than linear). We used the same 0.4 dropout and 0.001 learning rate. We also changed the 270-day look back window to a 365 day one to better account for the snow processes in Canada. The results show that the median testing KGE over all catchments was 0.71 (vs 0.77 for our model with no-donor catchments, and 0.82 with donor catchments). During the development of the models in the paper, we did this process with increasingly complex models and ended up with those used in the paper. We built upon this experience through various tests to provide this present model, even though it is indeed quite large and complex. But this is still a small model compared to models such as AlexNet and LLMs, and we believe that the addition of more data will always require more complex models to make use of it as best as possible. The figure below shows raw data for the various runs as in those in the "major modification" section at the top of this document, with the simple model runs in the last column for comparison.

[Figure]

**Figure RR1**: This figure is the same as Figure 3a in the original paper, with the exception of the seventh boxplot, which corresponds to the modified continental model (with donor catchments (WD), excluding latitude and longitude (NLL), and no static climate descriptors (NS)), and the eighth boxplot adding the simple LSTM model described above. This model was only trained once and it is quite probable that performing multiple trainings would improve results to a certain extent, but not to the point of competing with the larger models.

3) Chapter 4.4.1.: results and literature stand next to each other disconnectedly; no true conclusion is drawn in the chapter. Insert it.

Thank you for this suggestion. Based on the recommendations from all reviewers, the manuscript has been restructured. This section was revised, and a clear conclusion (that also addresses other issues raised by yourself and other reviewers) has been reworked in the new version of the paper.

4) Results and discussion are formally separated from each other, but then there is another bunch of analysis and four (!) figures in the discussion section, which renders the separation of results and discussion irrational. Either clearly separate results from discussion, or – much preferred, because much easier to follow/ understand in general – make a single "results and discussion" section and discuss noteworthy point right next to the figures.

We agree with this point. In the revised manuscript, we combined the results and discussion sections into a single section, ensuring clear discussions aligned with the related literature. Please note that the changes are significant here and as such we do not provide more details on what was done exactly. Please see the track-changes version of the manuscript to see the full extent.

5) the conclusion is a stump and contains mostly commonplace statements. Revise to revolve around actual key conclusions from your results.

We have reworked the entire conclusion section to better reflect the key conclusions that are presented throughout the different results. The new conclusions also refer back to the paper's objectives as suggested by reviewer 3.

6) typos:

Line 576 "mpdels" à models

The typo "mpdels" has been corrected to "models".

Line 671 "streamflow electricity" à elasticity

The typo "electricity" has been corrected to "elasticity". The auto-correct was quite original on this one.

**Reviewer #2:** https://doi.org/10.5194/egusphere-2024-2133-RC2

The study aimed to compare process-based models (PB) and machine learning models (ML) when used for scenario analysis. Specifically, they assessed three common PBs and two Long-Short Term Memory (LSTM) configurations. The researchers focused on examining the sensitivity of streamflow to different forcing perturbations. Their findings indicated that LSTM trained at a continental scale is a more reliable model due to its training on a wider range of variability. However, the study suggested that in most cases, the sensitivity between PB and ML is similar, with a few exceptions.

Thank you very much for your comments and suggestions to improve the manuscript. Below, we provide point-by-point responses to all concerns, comments and suggestions.

Major comments:

1. The use of latitude, longitude, and climatic static attributes in the LSTM models restricts the model to learn local information rather than capturing the spatial variability in the dataset. This could affect the model's ability to accurately analyze sensitivity under different climatic conditions. I suggest running the model without latitude and longitude and using the minimum number of climatic attributes while changing them according to the sensitivity being analyzed (ex. 20% increase in precipitation means a 20% of the mean annual precipitation).

This is a good observation. We were curious about this when reading your comments and have retrained the model without using the latitude and longitude (or climate-based indicators) in the variables. Please see the "major modifications" section at the top of this document for an in-depth response to this point and how the paper has been modified to reflect this.

2. There is an excessive number of figures and sections presented. Some results are repetitive, and certain sections may not add significant information due to high uncertainty (ex. 4.4.2). I recommend including only the figures and sections that directly support the main conclusion of the report while considering moving additional figures and analyses to the appendix or supplemental information.

Based on comments from Reviewers 1 and 3, we have reworked and combined the results and discussion sections and have move some of the figures into the supplemental information. The complexity of the Figures has also been reduced. The original paper contained 12 Figures (many that were complex, e.g. 10 and 12). The new version now has 10 Figures, and a more streamlined presentation of Figures.

Minor comments:

Line 14. The comment about traditional hydrological models relying on historical climate data is misleading. ML models rely on historical data too.

Thank you for pointing this out. The sentence was reworded in the revised manuscript as follows: "*Traditional hydrological models, which rely on simplified process parameterizations with a*

*limited number of parameters, are scrutinized for their capability to accurately predict future hydrological streamflow in scenarios of significant warming.*"

Line 85. The definition of what is long and short is subjective. In fact, how long the memory in an LSTM model remains an open question.

This is a good point. The sentence was reworded to "*Their unique architecture enables them to learn and remember over longer sequences of data compared to RNNs, making them highly effective for predictions of time series*" in the revised manuscript.

Line 109. The problem is not the new climatic scenario, the problem is how to define when LSTM is in extrapolation mode. Because we can have an extreme condition in one catchment but the same would be normal in another, so the model can infer the relationship. In that case, the model is still in interpolation mode.

Agreed. This was reformulated in the revised paper. The advantage of using all of these extra donor catchments is to widen the training dataset characteristics such that the climate model-driven simulations are mostly in interpolation mode rather than extrapolation.

"*This assumption may not always hold, especially under scenarios of unprecedented climate change, however, the inclusion of a wider spatial sample of watersheds could provide the LSTM model with enough information so it stays in interpolation mode.*"

Line 118. You should have an introduction between sections and subsections. Probably a title as a dataset or data would match better with what you have.

We have added a short introduction between sections and subsections. However, we do not understand the second part of your comment. The section is titled "Study area and Data", as we describe the study area in 2.1 and the data in 2.2.

Line 132. More data is better, but how did you define this number?

It was not possible to take more than approximately 30 years of data without having to drop a significant number of watersheds. This is now mentioned in the revised manuscript as follows:

"*Catchments also required at least 30 years of data over the 1979-2018 period to be selected in order to have sufficient data to train both the conceptual hydrological models and the deep-learning implementations. The basin selection criterion was set to a minimum of 30 years of data to ensure not only a sufficient data length but also a robust sample of basins for performance assessment. These criteria were also used in this study, resulting in the selection of the same 148 catchments for the analysis.*"

Line 161 – 166. You do not need that paragraph. You can use a reference to explain this in more detail.

We have made the paragraph significantly shorter and added the following reference: *Wickham, H., & Stryjewski, L. (2011). 40 years of boxplots. Am. Statistician, 2011:*

*"Boxplots will be used throughout this paper to outline study results (Wickham and Stryjewski, 2011). A boxplot is a concise graphical tool which highlights the central tendency (median), variability (interquartile range; IQR), and outliers (data extending beyond the whiskers or 1.5 times the IQR) within the distribution of results across all of the study catchments."*

Figure 1. It needs a legend.

We have added a legend explaining the difference between study catchments and donor catchments (large circles with black outline vs small circles with gray outlines for the a and b panels, and green and orange boxes for the c and d panels respectively). Please see the new figure 1 here for your convenience:

[Figure]

Figure 1: Maps (a, b) of study area showing the location of the 148 studied catchments (large circles with black outline), and the 1,000 donor catchments for the continental LSTM model (small circles with grey outline). The fill colour represents the mean annual temperature (a) and total precipitation (b) of each catchment. The circles are located at the centroid of each catchment. Box plots showing the comparison of mean annual temperature (c) and total precipitation (d) across the target sample of 148 catchments (green boxes) and the 1,000 donor catchments (orange boxes) for the continental LSTM simulation.

Line 193. This attribute shouldn't be used because you are anchoring the dynamic to a location which is exactly what you are trying to avoid. That can have serious effects on the sensitivity of your model. AND Line 195. How are you disentangling the correlation between catchment attributes and the meteorological forcing?

These two comments are related to the major comment related to the latitude and longitude statics as well as the climate-based static descriptors. Please see the major comment at the beginning of this document for a detailed response to this point.

Line 317. How did you apply that modification? only testing or in the entire period?

This was done over the entire period. This has been specified in the revised manuscript as follows:

*"To achieve this, historical weather data over the entire period was modified by applying predetermined factors to create new datasets that served as rough estimates of future weather conditions."*

Line 404. This is not the reason for not presenting the validation period. All your results should be in a period that was never used during the training and validation.

This is correct. We reformulated as follows: "*Note that the validation period for the LSTM-based models is not shown as the validation data are contaminated by the training data, and thus, should not be investigated.*"

Figure 3. I recommend using CDF plots. This format has been widely used in streamflow models. I suggest adding a line where the best value is found.

We tested using CDFs, but we found that the graphs are much easier to compare results together using boxplots. Thus, we have decided to keep boxplots instead as the main tool for analyses.

Line 453 – 454. This could be a consequence of using latitude and longitude as input. The LSTM model is fixed to the location so it is less sensitive to precipitation because part of the precipitation correlation is shared with those attributes or with the climatic ones.

This is a good comment and is now resolved in the revised version. Please see the main major comment at the beginning of this response document for information on how this was handled.

Line 461 – 462. Something similar could be happening here with elevation. Temperature and elevation are very correlated. An interesting experiment would be increasing temperature and decreasing elevation by using an altitudinal gradient. Should the sensitivity increase or decrease?

See Supplemental material Figure S1b) (of the revised manuscript) to see the distribution of elevation across all watersheds. All watersheds are within a 500-meter range more or less, thus the impact of elevation on temperature should be marginal, much less so than the impacts of climate change.

Figure 4. I would decrease the y-axis range. Probably [-60,20] for a and b. [-50,50] for c and d.

The y-axis ranges in Figure 4 (and now Figures S4 to S8 in the supplementary materials) were modified to reduce the white space and to improve the display of the results. Among a group of four figures, the same range was kept, allowing for comparison (e.g., -10 to 40 is the same range as 0 to 50).

Figures 5 and 6. Move it to the appendix or supplement information.

We have substantially modified the general structure of the manuscript based on all reviewer comments. Please refer to our reply to your main comment 2 for further details.

Line 479 – 480. Explain why it is clear to you.

This is based on Figure 2 which shows mean annual temperature and precipitation changes for each of the 22 GCM models. Median temperature increases range from +4 to +8.5 °C and from +4 % to +20 % for precipitation. Based on this, the +6 °C and +20 % sensitivity scenarios are a lot more realistic than the +3 °C and -20 % ones. We have rephrased and clarified this statement as follows in the text:

*"Figure 2 shows that median temperature increases range from +4 to +8.5 °C and from +4 % to +20 % for precipitation. Based on this, the +6 °C and +20 % sensitivity scenarios are more realistic than the +3 °C and -20 % ones."*

Figure 7. In many cases the differences look not significant, you should do hypothesis testing to check the level of significance. Moreover, you should mention something about the differences in the variability between some models. All the sub-figures must have the same y-axis range to do a fair comparison. I do not think you need all the sub-figures; you should show here only the most significant.

Good point for the figures. We have rearranged Figure 7 (now figure 5) so that the range (maxY - minY) is the same for all 6 panels, except panel c) due to the widely different y-axis values for that case (as shown in the caption). We also double-checked other similar Figures for the same issue. We performed statistical tests, the results of which are now presented just before figure 5. More specifically, the nonparametric rank sum test for equality of medians was used. The tests showed that the LSTM results were in almost all cases statistically significantly different from the process-based models. We have now expanded on this in the revised version, with the following text:

*"A Wilcoxon signed-rank test was used to test for statistical differences between the LSTM-based models and the conceptual hydrological models. LSTM-based models are statistically different than all other models in all cases except the following:*
- *Figure 5a) (QMA): LSTM-R is not statistically different from HSAMI and LSTM-C is not statistically different than GR4J.*
- *Figure 5e) (QMJJA): LSTM-R and LSTM-C are not statistically different from HSAMI."*

Line 537. No result can be considered a result, but in that case, you could move the entire section to supplemental information.

In this case, we prefer keeping this analysis here as it also supports the fact that the LSTM-based models and the classical hydrological models are providing similar responses to the same inputs. The fact that the results show no difference is actually very positive for the analysis. We therefore kept this as-is, albeit in the new "results + discussion" section as discussed in another set of comments.

Line 558. But exactly for this reason you have the third period. Are you trying to say that this period is not representative enough? If this is the case, you should check that.

This sentence aims to convey the fact that in typical hydrological modelling, we have 2 phases: calibration and validation. But in Deep Learning, there are 3 phases required: Training (equivalent to calibration), validation, and testing (equivalent to validation for hydrological models). There is no equivalent of the "DL-Validation" in the hydrological modelling world. This data is used as a stopping criteria only, and is not used to define the scaler parameters nor is it used as training data to determine the model weights. It is used to stop the training process once the best validation is achieved, to prevent overfitting. Without this, models would converge to near perfection in training but would be completely useless in testing due to the massive overfitting. This has been clarified in the text by removing this sentence here but clarifying the methods section.

Line 559 – 561. This is not part of the discussion; this is just part of the methodology.

We have removed this statement from the discussion and incorporated it into the LSTM model description in the revised manuscript.

Line 563 – 564. This is a strong statement without support. Remember that the number of parameters is not comparable between PB and ML models.

The vastly different number of parameters is what is referred to in the term "significantly greater number of degrees of freedom" in the original text. We rephrased the sentence as follows: "*From Figure S3, it is evident that LSTM training period performance surpasses that of conceptual models.*" and removed the notion of degrees of freedom.

Line 568 – 569. This sentence is exactly the opposite you said in line 558. You should put everything in one paragraph to tell a more consistent story.

Yes, the sentence is opposite to the previous one because the first one refers to the training period (i.e. calibration) where the performance can be essentially perfect with a large enough model. We state that the training performance should not be analyzed deeply because it is essentially unusable to verify model performance, contrarily to traditional hydrological models. In lines 568-569, we are talking about the independent testing performance, which can be used to assess model performance and where no signs of overfitting are seen. This is due to the precautions taken to

prevent overfitting, using the validation period (the DL-validation, not the validation of the hydrological model) as a stopping criteria for training. We have rewritten this section to better clarify the intent along with clarifying some points of the other reviewers. We recommend looking at the revised text in section 4.4 as the changes are too numerous to indicate here.

Line 571 – 572. 1000 sounds like a large number of catchments so I disagree.

This sentence simply states that we saw an important improvement in model results when going from 148 to 1148 (+1000) catchments, and thus this confirms that adding a large number of catchments improves performance. We do not understand what this comment is referring to that could be disagreed with or interpreted differently.

Line 575 – 576. Are you talking about distributed models? If this is the case, this would not be a fair comparison. If you want to add more degrees of freedom to PB, for example, you could combine the different sources of precipitation. Moreover, remember that the parameters in a PB encode the local descriptors (local characteristics) that the model needs.

It is true that we could use more sources of data to generate ensembles for example, which would allow averaging using a model averaging system that could indeed add more degrees of freedom. However here we meant that a single model simulation of a PB cannot make use of multiple precipitation datasets at once unless that model was specifically built to handle multiple precipitation inputs at once. The structure is fixed and cannot be increased. LSTM models afford this capability. And yes, the PB models encode the information directly into the model structure, but this also means that these are fixed in space and time, whereas an LSTM model can be applied to an ungauged location by using the descriptors, and can add more if needed. Ex. some PBs do not use the soil porosity or hydraulic conductivity explicitly, whereas an LSTM model could add one with no issue as long as the data is available.

Line 580. I disagree. More variables increase performance but decrease interpretability. How could you do the same sensitivity analysis with 10 hydroclimatic variables highly correlated?

We agree with this, the interpretability would be much lower with many highly correlated variables. But this is not what we are advocating. We are expressing that adding mode data could (or most likely would, based on other papers) increase model performance, and could make the model more likely to respond accurately to climate change signals due to the extra information. A model using only precipitation would definitely see improvements in adding temperature. We are simply extending this reasoning to higher orders.

Line 590 – 591. You do not need to be sorry for finding that those models are the worst, this is just part of the results. Delete the sentence.

This sentence has been deleted.

Line 605 – 610. I disagree. The situation is exactly the opposite. You are not counting all the sensibilities; a temperature change can change the precipitation too, meaning that the final

sensitivity of streamflow can be higher or lower. So, your analysis is a simplified sensitivity analysis which does not mean you are more accurate.

*We certainly did not mean to imply that varying one-parameter at a time is better despite our poor choice of words. Of course, a temperature increase definitely affects precipitation in a myriad of different ways (mean, variability, extremes). Simple sensitivity analysis methods typically vary one parameter at a time, and we still believe that this is a useful approach (as stated in the paper) to better understand the potential difference in sensitivity between the classic hydrological models and the LSTM class of models.*

*We have rephrased the original sentence: "By independently altering each variable—precipitation and temperature—we were able to more accurately evaluate the impact of each change…"*
*with the following:*

*"By independently altering each variable—precipitation and temperature—we were able to quantify the impact of each change, thus avoiding the complication of introducing confounding factors, which is the case when solely relying on GCM simulations for this analysis"*

Line 622. What family is that?

*We have rephrased to: "The four traditional hydrological models share similar structures, all being lumped conceptual models".*

Line 625. This is a contra argument about sensitivity coming from structure.

*We are not entirely sure what this comment applies to. This sentence aims to show that the sensitivity was similar across the 6 models, including 4 conceptual ones with similar structures, and 2 LSTM-based models with extremely different structures. Thus, we do not think the sensitivity comes from the structure, but from the meteorological inputs.*

Section 4.4. You focus just on which is the best. You must analyze the benefit of the ensemble of models (multi-representation approach). The concept of the best model does not exist.

*While a multi-model approach can be used to provide a better evaluation of the overall uncertainty, our results support the conclusion that when using a single hydrological model, the LSTM-based model is likely to provide a better evaluation of the impact of climate change on the hydrological process. However, we agree that adding a section in the text highlighting the potential of a multi-model approach to evaluate the overall uncertainty is important, thus we have done so, as follows:*

*"This is ultimately the most important question, but also the most difficult one to answer. Since there are no future streamflow data available, we are mostly left with theoretical arguments. One way to navigate this issue is simply to state that these LSTM models should be included in multi-model ensembles to better assess modelling uncertainty (e.g., Dams et al., 2015; Najafi and Moradkhani, 2015). However, many studies suggest that we should have more confidence in hydrological models that yield better results in the historical period, as they are better at*

*representing processes (e.g., Krysanova et al., 2018), which leads back to the initial question: Which type of model should we trust more?"*

Also, we have reformulated the section heading as follows: "Which type of hydrological model should we trust more for climate change impact studies?"

Line 653. That is not true. There is a lot of research on interpretability. We do not have yet the same level of interpretability as PB, but this does not mean we are not going to get it in the future.

This is a correct assessment. For some simple deep learning models there is starting to be some progress in interpretability, and the statement has been changed to clarify that this is not realistic at the moment but that with time this might be something that becomes possible. The text now reads:

*"However, LSTM model process representation is quite challenging and obfuscated, meaning it cannot easily be probed directly to assess how the hydrological cycle is modeled. This is a limitation, as it is currently very difficult or nearly impossible (depending on the model complexity) to assess this in an LSTM-based model, which means it requires more blind trust than conceptual models."*

Figure 9. I would prefer a table or a CDF figure showing the distribution. It is very hard to compare models within each group.

As stated for reviewer 1, we tested using CDFs, and the quality and readability was worse than using boxplots. We therefore opted to keep boxplots.

Line 701 – 703. I agree about the catchment attributes used however you considered only input similarity. This is not enough to define dynamic similarity, at least you should consider adding similarity in the streamflow signature.

We understand the reviewer comment. However, this is not possible for this assessment. The reason is that we are trying to see if the simulated flows under a future climate are well represented by the LSTM model. To do so, we need to find indicators that are independent of streamflow, as using a streamflow signature metric to find an analogue would ensure that the catchments found had similar hydrographs. This would defeat the purpose of ensuring that the analogues in terms of climate are indeed analogues in terms of hydrology.

Line 705. If this is the case, you should use a uniform weight distribution. Add information supporting your decision to use something different than uniform.

We have completely changed the section on analogues. Instead of trying to find analogues for the +6 °C and +20% sensitivity scenarios, we have redone the work for the 22 climate change scenarios instead. The problem with the +6 °C and +20% sensitivity scenarios is that it was difficult to find proper analogues since precipitation and temperature are somewhat related. This is why putting more weights on T (for the +6 °C scenarios) or P (for the +20% scenarios) yielded slightly better results. For the climate change scenarios, equal weighing was used. In addition, using GCM

provide more realistic climate projections. Ultimately, this does not change the conclusions drawn in the first version of the paper, but it provides more robust results.

Figure 10. Given the huge difference between the analogues and the models, it is impossible to say that one model is better than the other. Moreover, if I suppose that the catchments presented are the best ones, it is impossible to infer more from this type of comparison.

Analogue analyses are, by definition, uncertain. We provide this analysis with the stated aim of finding adequate analogues (10) for each case to encompass the uncertainty of selecting the "best" analogue catchments. Looking at the hydrographs, it is clear that the models are all representing the hydrological cycle generally correctly, albeit with some key differences in select aspects of the hydrographs. We can definitely see that LSTM models perform better in some cases and classical models perform better in others, which is expected. However, it is clear that the LSTM based models are not consistently worse in any part of the hydrograph, but present a balanced comparison with the hydrological models. This is reassuring, in that the objective of the paper is to investigate if LSTM-based models can be used for climate change impact studies, and this analysis seems to confirm that they are at least not worse than classical hydrological models.

Line 759 – 764. Exactly for this reason I would drop the entire section. The results from this section are not different than before but with a huge uncertainty.

We respectfully disagree about dropping the entire section. While there are indeed some limitations to this analysis, it still provides valuable insight on the ability of the different types of hydrological models to simulate streamflow under a changing climate. It is one of the only insightful methods to assess the ability of models to represent climate change and to validate those results. Furthermore, these results are supported by the other results from this paper and thus contribute to the overall objectives of the paper. There is definitely some uncertainty, but this does not warrant, in our opinion, removing the analysis and its findings.

**Reviewer #3:** https://doi.org/10.5194/egusphere-2024-2133-RC3

The paper evaluates LSTM-based hydrological models and traditional hydrological models in Climate Change impact assessments. The models are assessed regarding their ability to simulate future streamflow and against their sensitivity to climatic changes. The authors conclude that LSTMs, given they are trained with sufficient data, are a viable and most likely a better alternative for climate change impact assessments.

Thank you very much for your comments and suggestions. We provide point-by-point replies to each of your comments below.

In my opinion the title does not fully express what you actually did in the study. You assessed the adequacy of both traditional models and LSTMs. Not sure, you might want to highlight that.

We agree that we assessed both lumped traditional hydrological models and LSTMs through the proposed methodology. However, our main goal was to highlight the potential effectiveness of LSTM neural networks compared to traditional hydrological models. Therefore, we consider that the current title reflects the intended take-home message for the readers.

The authors address an important topic and use a suitable dataset and methods for the evaluation. However, the main problem I have with the paper is the structure:

Nowhere in the paper do the authors refer back to the three objectives outlined at the end of the introduction. I would expect that you explain how you are going to achieve the objectives in the methods, show the respective results and discuss those, and ideally come back to the objectives in the conclusion. The paper seems unstructured in that regard and it is hard to understand which of the subchapters contributes to which objective, thus disconnecting the analysis from the objectives.

This is a valuable point also raised by Reviewers 1 and 2. Therefore, following the reviewers' comments, we have restructured the paper by combining results and discussion sections together. We have moved some of the Figures in the supplemental materials, and simplified others. The three objectives are now referred back to in the reworked conclusion.

Another structural problem is already evident in the abstract: Your last sentence of the abstract highlights the analysis of precipitation elasticity and catchment analogues. I like these two analyses, but I find it strange that these are shown (including figures) in the discussion only. I do not see a reason why these analyses should not be structured into methods, results, discussion.

This is a good point, and we have addressed it by combining the results and discussion sections, as mentioned in the previous reply.

Additional Comments:

l.130-134: It seems unclear at this point if the current study is also limited to >500km² and 30yr data. Suggest to add a brief explanation, particularly as you mention later (l.159) that you restrict to 20yr streamflow data.

This has been clarified by modifying this paragraph as follows:

"*In the Arsenault et al. (2023) study, only those catchments with a drainage area exceeding 500 km2 were included, thereby sidestepping potential issues related to scale and time-lag in model regionalization efforts. Catchments also required at least 30 years of data over the 1979-2018 period to be selected in order to have sufficient data to train both the conceptual hydrological models and the deep-learning implementations. The basin selection criterion was set to a minimum of 30 years of data to ensure not only a sufficient data length but also a robust sample of basins for performance assessment. These criteria were also used in this study, resulting in the selection of the same 148 catchments for the analysis*".

The 20-year limit was set only for the extra 1000 donor catchments to widen the set of available catchments for this analysis, but these were not as critical as the original 148 as they were not used for model testing. Therefore, this constraint was relaxed to 20 years for the extra set of catchments. This has been clarified in the text as follows:

*"Note that the 20-year limit differs from the 30-year limit used for the study catchments selection to widen the set of available catchments for this analysis, but these were not as critical as the original 148 as they were not used for model testing. Therefore, this constraint was relaxed to 20 years for the extra set of catchments."*

l.136: I think the term 'scenario' is somewhat misleading in this context. I suggest to write something along the lines "An extra set of 1000 donor catchments was selected for an additional LSTM application."

Thank you for your suggestion. The term "*scenario*" has been replaced with "*LSTM configuration*" as suggested in the revised manuscript.

l.149: against the background of the previous sentence, it is unclear what you mean by 'common denominator for all models'

We clarified the author's meaning by adding, "*(i.e., the one that corresponds to the intersection between both datasets)*" in the revised manuscript.

l.153-154: suggest to check Tarek et al. if that is generalizable for any catchment / region.

Thank you for your suggestion. The study presented by Tarek et al. was done for all North America, covering our entire study region. The study shows indeed a generally good performance over the whole region compared to observations, with some loss in performance in eastern U.S. compared to observations. Nonetheless, the performance remains satisfactory for the present study.

l.195: I assume the climatic descriptors are kept constant under climate change? Do you think considering a possible spatial shift of these climatic conditions under climate change could further improve the models? Perhaps a point worth discussing.

Please see the major modification presented at the beginning of this document for more information on how this was resolved and implemented in the revised version of the paper.

l.255: μ is missing in explanation

We have added that μ refers to the average of the simulation and observed streamflow, respectively, in the revised manuscript:

*"where σ represents the variance, and $\mu_{sim}$ ($\mu_{obs}$) the average of the simulation (observed) streamflow."*

l.289-290. I do not understand this part. Why is data combined from multiple catchments for the computation of the objective function? And why was the NSE used for the LSTM's objective function and not the KGE as for the classical hydrological models?

Since the model is a regional one (or continental for the one with the extra donors), it requires being fed data from various catchments. The order that these data are fed into the model for training is randomized to ensure stochasticity and to help convergence. Therefore, each batch needs to compute an objective function on a subset of the data sampled at random from hundreds of thousands of data points. To make this possible, the objective function needs to allow this. The objective function is an NSE-based metric that is slightly altered to ensure that the flow values are scaled appropriately, or else the larger catchments would weigh more than the smaller ones simply due to their size (and thus larger flows). Doing so with KGE is more complicated because the ratio of variability is highly impacted by the very small sample sizes of the batches. However, for evaluation, using KGE makes more sense as all flows are available, and the metric allows for more in-depth evaluation. For the traditional hydrological models, we would not use the same NSE-derived and scaled objective function as it is done one catchment at a time. Thus, we simply decided to use KGE directly. This means that the classical hydrological models are given a bit of an advantage over the LSTM models, but this is minor and, as the results show, the models are still comparable over the independent validation/testing period.

We have clarified all of these points in section 2.3.2, and we refer the reviewer to that section for a list of all changes as they are too numerous to indicate here.

l.305: Suggest to introduce LSTM-R and LSTM-C earlier in the manuscript and mention the respective 148 and 1000 catchments.

We have now introduced LSTM-R and LSTM-C and the respective 148 and 1000 catchments configurations earlier in the manuscript as suggested.

l.322-327: I assume when implementing one test, all other variables were held constant (so no combination of TMP and PCP changes)? I suggest to mention that.

Yes, exactly. We have now clearly stated in the revised manuscript that no combination of temperature and precipitation changes was used in any of the tests. Changing both variables at the same time complicates the interpretation. The 22 GCMs provide projections where both variables change.

*"Note that no combination of temperature and precipitation changes was used in any of the tests."*

l.384-385: why no low flow metric, such as the annual minimum streamflow? Also, I suggest to add the time periods for which these metrics are calculated (I assume the hindcast and the future climate?)

In this work, we have computed and analyzed 51 different streamflow metrics. These metrics cover mean flow values, distribution quantiles, and low- and high-flow extremes. We chose to focus on mean annual and seasonal streamflows, as these are robustly simulated by process-based hydrological models and serve as key climate change indicators. Low flows present challenges for process-based models, with a high level of uncertainty in projecting low flows in a warmer climate, where hydrological model uncertainty is much larger than that of GCMs. We also included the mean of maximum annual streamflow, which, as a "moderate extreme," is well represented by hydrological models. More extreme low- and high-flow metrics would require additional validation, especially for LSTM models. The following sentence was added in the revised manuscript:

*"In addition, the six streamflow metrics were chosen as they are all reliably simulated by all 6 hydrological models over the reference period. Low flow and large extremes metrics were not chosen as they are less reliably simulated by the hydrological models over the reference period, and therefore, projecting these metrics in the future comes with larger uncertainty."*

l.395-397: You mention NSE and KGE at three locations now. I am confused what data and models are used for which metric. And if different metrics were used for different models, I see a significant bias here given that you compare other metrics (see 2 comments further below). I suggest to mention the KGE and NSE metrics only once in the manuscript to avoid confusion. Also, in l.403 you mention NRMSE which was not mentioned in the methods.

Considering that the objective function used to train the LSTM model incorporates all catchments at the same time, it is necessary to add a standard deviation weighting to avoid overfitting catchments with larger streamflow. No identical objective function can be fully used for both types of models. In all cases, the traditional hydrological models are favored by this methodological choice, as mentioned previously (section 2.3.2.)

We also added the NRMSE equation in the method section to avoid any confusion:

*"Finally, a third and final "general" metric that is independent from the KGE and NSE, used for training the conceptual models and LSTM models, respectively, was implemented. This metric is the Normalized Root Mean Square Error (NRSME) and is the RMSE normalized by the range of the streamflows in the timeseries. This allows comparing results between watersheds despite their size differences. It is computed as follows:*

$$NRMSE = \frac{\sqrt{\frac{1}{n}\sum_{i=1}^{n}\left(\left(q_{obs}^{i}-q_{sim}^{i}\right)^{2}\right)}}{max(q_{obs})-min(q_{obs})} \qquad (2)$$

*where $q_{obs}$ and $q_{sim}$ are the observed and simulated flows, respectively, and n is the number of days of data in the evaluation period."*

l.410: You did not introduce the optimum values for each metric. I suggest to add this information to where the metrics are introduced first, or/and the optimum could be added as a line to the diagrams.

We have added the optimum values for each of the metrics in section 3.1 of the revised manuscript as suggested.

*"The optimal value for each metric is as follows: KGE = 1, NSE = 1, β = 0, r = 1, γ = 1, and, NRMSE = 0."*

l.431: Could the difference you see in the variability ratio between conventional and LSTM models be due to the different objective functions you used (KGE vs NSE)? The tradeoff between the different performance criteria is interesting. For further discussion of the relationship between the performance criteria, you can look into Guse et al. 2020 (https://doi.org/10.1080/02626667.2020.1734204)

This is a good idea. We analyzed the impact of the objective function on the variance ratio for the various models and detail our findings in the text as follows:

*"This could be related to the objective function used, as the LSTM training operated on a metric that inherently scaled simulated and observed streamflow by the standard deviation of the observations during its computation, lowering its impact. Conceptual models were calibrated individually using KGE, which has a term specific to variance and is thus directly considered. However, this should favor the conceptual models having better (i.e. closer to 1.0) values of the variance ratios."*

l.442-443: suggest to summarize the main results of the other metrics here. I would assume the QMM is of interest to many readers.

We have modified the text to better reflect results presented in the main paper as well as those presented in the supplementary material. Please see the revised "Results and discussion" section to see how this was done throughout the text.

l.465: I do not understand why a more pronounced response to temperature changes implies providing more accurate projections. Without comparing this to observations, I think this is not defendable. Please explain.

We agree with your comment. This was definitely worded incorrectly. The paragraph has been modified and the reference to more accurate projections removed:

*"Despite some variability between the different streamflow metrics, and across seasons, LSTM models show a different climate sensitivity than that of traditional hydrological models. They are less sensitive to precipitation changes across all metrics. The best performing LSTM-C model also shows a decrease sensitivity to temperature compared to the 4 traditional hydrological models, with the exception of fall (SON) flows. LSTM-R increased sensitivity to temperature at the annual scale (Figure 4) is largely the result of its very large sensitivity for the fall (SON) season (Figure S7). Its sensitivity tracks that of LSTM-C for the other seasons."*

l.500-501: I am not sure about this statement. The classical hydrological models project an increase in streamflow. That would mean precipitation is overruling temperature.

We have removed the statement.

l.509: what do you mean by "near the surface"?

"Near the surface" refers to a height of 2 meters above the surface. This was clarified as "near the surface (2 meters height)" in the revised manuscript.

l.532: Why did you evaluate this section only for QMA?

This paper is already too long (as mentioned previously by the reviewers). We have decided to select this variable to conduct this analysis. However, all data are available in the data repository and readers could evaluate the metrics of their choosing to determine if the model is appropriate for their needs. Internally, we analyzed 51 metrics (The 51 from Table 2 in Arsenault et al. 2020) and results are fairly consistent. We decided to only show QMA for space considerations, given that it is one of the main metrics used to evaluate impacts of climate change on hydrology. Had we gotten strange and unexplainable results for QMA, it would have been a clear sign that the LSTMs were perhaps not suited for climate change studies. Doing more metrics would be desirable but we must balance the amount of data and results presented with the overall message, we wish to expose.

Arsenault, R., Brissette, F., Chen, J., Guo, Q. and Dallaire, G., 2020. NAC2H: The North American climate change and hydroclimatology data set. Water Resources Research, 56(8), p.e2020WR027097.

l.561: I do not understand what you mean with "to prevent contamination"

This refers to "data leaking", meaning that any data from the testing or validation sets cannot be used to help build the scalers or be used in the process of defining the model weights. Doing so would let the model include that data during training and thus make it more robust in testing, meaning that the model would be "cheating" by accessing "future" data it should know nothing about. This has been clarified in the paper.

l.576-577: Why do you write here that the LSTMs only use the climatic data? I suggest to add "..rain, and snow besides the catchment descriptors represent a fraction...".

Good point, the descriptors have been added to this list in the revised paper.

l.582-585: There are many hydrological models around that can make use of additional physical and temporal data. While I understand the advantage of complex LSTMs that can ingest all this data, I think you should not overstress this 'advantage' only because you chose traditional lumped hydrological models only, that cannot use additional data.

Actually, this section of the text points to a limitation of the LSTM models, in that while they can use a large variety of input data, that data also needs to be available for any simulation, forecast, or projection. This is a limiting factor that other models do not have (i.e. you could calibrate a hydrological model using remote sensing data to tweak parameters and then feed climate data to the model with no issue, but this is impossible with the LSTM-based models used here). Therefore, we left the text as is. As to the comment related to the ingestion of data, we have also clarified elsewhere in the text (see one of the comments below) that other hydrological models can use more sources of hydrometeorological data than those used in this study.

l.625-629: Isn't it also reassuring that the different classical hydrological models performed similar? It could also mean that the projections can be considered robust.

This is a good point, and this was added to the text:

*"All four traditional hydrological models were similar in their climate sensitivity, and even the simple MOHYSE model was not an outlier for the six streamflow metrics considered, which tends to demonstrate the robustness of the results."*

l.634-635: Well, this is clearly beyond scope for your study, but you could at least discuss that there are studies that used historical streamflow change to evaluate models (see for instance Eyring et al. 2019, https://doi.org/10.1038/s41558-018-0355-yor; Kiesel et al 2020, https://doi.org/10.1007/s10584-020-02854-8 who both propose out-of-sample evaluations). And also, Krysanova et al (2018, already cited in your manuscript) provide a 5-step validation procedure that allows an assessment of how well models are suitable for climate change impact assessments. This might also be worth discussing.

We've added a short paragraph to discuss the suggestion of using reference streamflow to assess hydrological model fitness for climate change impact studies. We did not include the Kiesel et al.

(2020) and Eyring et al. (2019), as these specifically targets climate model fitness, which is clearly outside the scope of this paper. This is the added paragraph:

*"However, beyond theoretical arguments, there are ways to practically investigate hydrological model fitness. Several authors (e.g. Krysanova et al., 2018; Roudier et al., 2016; Bérubé et al., 2022; Todorović et al., 2022) have suggested approaches using historical data to assess hydrological model fitness for climate change impact studies. While these approaches offer interesting insights, hydrological variability over the recent historical record remains small compared to expected changes by the end of this century (Bérubé et al., 2022) and it is therefore extremely difficult to assess the true sensititivity of hydrological models to climate change simply using historical data. We therefore looked at two different ways of assessing hydrological model sensitivity: precipitation elasticity of streamflow and climate analogues."*

Added references:
- Roudier, P., Andersson, J. C. M., Donnelly, C., Feyen, L., Greuell, W., and Ludwig, F.: Projections of future floods and hydrological droughts in Europe under a +2°C global warming, Climatic Change, 135, 341-355, 10.1007/s10584-015-1570-4, 2016.
- Bérubé, S., Brissette, F., and Arsenault, R.: Optimal hydrological model calibration strategy for climate change impact studies, Journal of Hydrologic Engineering, 27, 04021053, 10.1061/(ASCE)HE.1943-5584.0002148, 2022.
- Todorović, A., Grabs, T., and Teutschbein, C.: Advancing traditional strategies for testing hydrological model fitness in a changing climate, Hydrological Sciences Journal, 67, 1790-1811, 10.1080/02626667.2022.2104646, 2022.

l.646-650: I do not completely agree with this statement based on your study, considering that other traditional models exist that utilize additional data sources as well. I suggest to add to l.648: ... a theoretical advantage over the four traditional hydrological models used in this study.

We agree to a certain extent, but we still believe the theoretical advantage holds. The advantage comes not from the fact that more data variables can be ingested (which other models can do, such as humidity, radiation, wind and other such variables). The advantage comes from the fact that the LSTM model ingests data from over 1000 diverse catchments and can learn how different climates lead to different streamflow values. No other model can do this, unless we count large-scale regional models that are calibrated on a large and diverse region. But even then, the parameters are fixed in time and the changed meteorological data is still fed to a deterministic model that cannot modulate flows according to nonlinear processes between meteorology and hydrology. This is where the LSTM strives. We therefore kept this in the text but better contextualized it, as done in this response. The modified text is as follows:

*"The inclusion of 1,000 additional catchments, mostly located in a warmer climate, indicates that LSTM-C should have, at the very least, a theoretical advantage over single-catchment or local models. It has learned the complex relationship between climate variables and streamflow, not only over the study domain but also from 1,000 catchments, many of which are representative of, and even warmer than, the expected end-of-century climate over the study domain. Other types of hydrological models (regional, distributed and more physically-based models) can also make use of some aspects of this trove of data (more catchments, more meteorological variables), but they*

*still face the problem of stationarity: Model parameters are still fixed in time and thus the model cannot account for non-stationarity in the same way the LSTM models can. LSTM models are particularly fit at capturing the complex, non-linear climate interactions leading to streamflow due to this large-scale training and exposure to various climates."*

l.660: Why is this a disadvantage of conceptual hydrological models? You also fixed the LSTM parameters after training?

Yes, but it is adaptive and has many internal weights able to process the various seasonalities within. This would be more similar to a hydrological model whose parameters are calibrated to different values for every day, for example. This was changed in the text as follows:

*"For example, the parameter sets of conceptual hydrological models are fixed during the calibration period with the hypothesis that these are constant over time (e.g., seasonally). However, this has been shown to be inaccurate (e.g., Kim et al., 2015; Mendoza et al., 2015; Bérubé et al., 2022), and LSTM-based models are not constrained to these same processes and can learn to modify streamflow patterns according to hydrometeorological conditions through their immense number of internal weights."*

l.685-694: in l.690, could you mention percentages instead of "a few" or "most"? Also, you could simply calculate the elasticity ratios yourself based on your the historical data?

We added quantitative values. Computing precipitation elasticity is fairly complex as the "devil is in the details". This would add significant length to a paper that was already deemed long by the reviewers.

l.773-777: I suggest to also mention that you did not include more physically-based hydrological models that can utilize additional data which could react differently to the future climate forcings.

We have added a sentence in the reworked conclusion about the potential advantages (and limitations) of physically-based models. We refer the reviewer to the revised conclusion to see how this was integrated in the conclusion.

Minor comments:
l.31 consider "studies evaluate" to avoid repeating "assess"
l.182 comprehnsive -> comprehensive
l.280 ...performance gains in other... ?
l.370 you express pcp change in %, therefore *100 should be added to the calculation example.
l.511: wet vs dry models
l.576: mpdels -> models
l.671: electricity -> elasticity
l.720: unit should be m3 s-1 km-2 ?
l.730 and supplementary material Figures: suggest to add or mention the RMSE unit of normalized streamflow.
All of these minor comments were corrected in the revised manuscript.